# Attractor Memory for Long-Term Time Series Forecasting: A Chaos Perspective

**Jiaxi Hu**[1], **Yuehong Hu**[1], **Wei Chen**[1], **Ming Jin**[2], **Shirui Pan**[2], **Qingsong Wen**[3], **Yuxuan Liang**[1] *

[1]The Hong Kong University of Science and Technology (Guangzhou)
[2]Griffith University  [3]Squirrel Ai Learning, USA
{jhu110, yhu322, wchen110}@connect.hkust-gz.edu.cn
{mingjinedu, qingsongedu}@gmail.com
s.pan@griffith.edu.au

## Abstract

In long-term time series forecasting (LTSF) tasks, an increasing number of works have acknowledged that discrete time series originate from continuous dynamic systems and have attempted to model their underlying dynamics. Recognizing the chaotic nature of real-world data, our model, ***Attraos***, incorporates chaos theory into LTSF, perceiving real-world time series as low-dimensional observations from unknown high-dimensional chaotic dynamical systems. Under the concept of attractor invariance, Attraos utilizes non-parametric Phase Space Reconstruction embedding along with a novel multi-resolution dynamic memory unit to memorize historical dynamical structures, and evolves by a frequency-enhanced local evolution strategy. Detailed theoretical analysis and abundant empirical evidence consistently show that Attraos outperforms various LTSF methods on mainstream LTSF datasets and chaotic datasets with only one-twelfth of the parameters compared to PatchTST. Code is available at https://github.com/CityMind-Lab/NeurIPS24-Attraos.

## 1  Introduction

In the intricate dance of time, time series unfold. Emerged from continuous dynamical systems [36, 11, 39] in the physical world, these series are meticulously collected at specific sampling frequencies. Like musical notes in a composition, they harmonize, revealing patterns that resonate through the symphony of temporal evolution. In this realm, Long-term Time Series Forecasting (LTSF) stands as one of the enduring focal points within the machine learning community, achieving widespread recognition in real-world applications, such as weather forecasting, financial risk assessment, and traffic prediction [28, 22, 30, 27, 18].

Building on the success of various deep LTSF models [49, 54, 52, 50, 29, 22], which primarily leverage neural networks to learn temporal dependencies from discretely sampled data. Currently, researchers [46, 32] have been investigating the application of Koopman theory [48, 25] in recovering continuous dynamical systems, which applies linear evolution operators to analyze dynamical system characteristics in a sufficiently high-dimensional Koopman function space. Nevertheless, the existence of Koopman space relies on the deterministic system, posing challenges given the chaotic nature of real-world time series data, evidenced through the Maximal Lyapunov Exponent in Appendix E.2.

In this paper, inspired by chaos theory [8], we revisit LSTF tasks from a chaos perspective: Linear or complex nonlinear dynamical systems exhibit stable patterns in their trajectories after sufficient evolution, known as attractors. As illustrated in Figure 1(a), attractors can be classified into four

---

*Y. Liang is the corresponding author. Email: yuxliang@outlook.com

38th Conference on Neural Information Processing Systems (NeurIPS 2024).

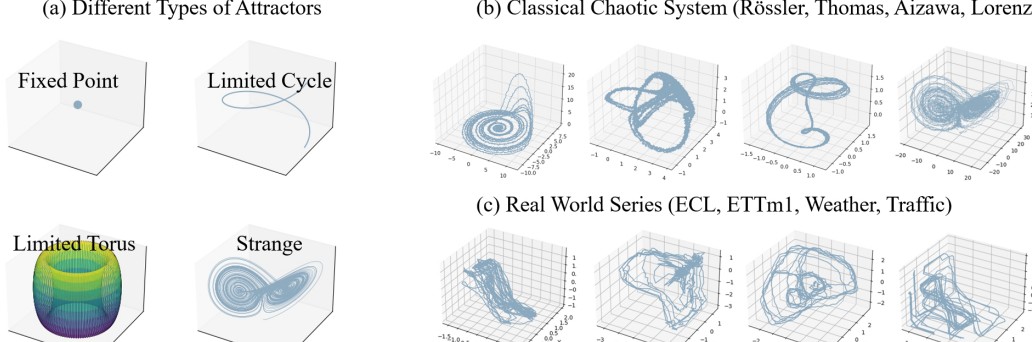

Figure 1: (a): Classical chaotic systems with noise. (b): dynamical system structure of real-world datasets. (c): Different types of Attractors. See more figures in Appendix E.1.

types: Fixed Point, indicating stable, invariant systems; Limited cycle, representing periodic behavior; Limited Toroidal, exhibiting quasi-periodic behavior with non-intersecting rings in a 2D plane, reflecting temporal distribution shifts; and Strange Attractor, characterized by nonlinear behavior and complex, non-topological shapes. Supported by chaos theory, we can transcend the limitations of deterministic dynamical systems to construct generalized dynamical system models. Figure 1(b-c) showcases various classical chaotic dynamical systems and the dynamical trajectories of real-world LTSF datasets using the phase space reconstruction method [9]. Notably, the dynamical system trajectories in these LTSF datasets exhibit fixed structures akin to those in typical chaotic systems.

Given this chaos perspective, we consider real-world time series as stemming from an unidentified high-dimensional underlying chaotic system, broadly encompassing nonlinear behaviors beyond periodicity. Our focus centers on recovering continuous chaotic dynamical systems from discretely sampled data for LTSF tasks, with the goal of predicting future time steps through the lens of attractor invariance. Specifically, this problem can be decomposed into two key questions: ***(i) how to model the underlying continuous chaotic dynamical system based on discretely sampled time series data; (ii) how to enhance forecasting performance by utilizing the attractors within the system.***

In this context, ***Attraos*** emerges with the goal of capturing the underlying order within the seeming chaos via attractors. For tackling the first question, we employ a non-parametric phase space reconstruction method to recover the temporal dynamics and propose a *Multi-resolution Dynamic Memory Unit* (MDMU) to memorize the structural dynamics within historical sampled data. Specifically, as polynomials have been proven to be universal approximators for dynamical systems [4], MDMU expands upon the work of the State Space Model (SSM) [12, 10, 18] to different orthogonal polynomial subspaces. This allows for memorizing diverse dynamical structures that encompass various attractor patterns, while theoretically minimizing the boundary of attractor evolution error.

To address the second question, we devise a frequency-enhanced local evolution strategy, which is built upon the recognition that attractor differences are amplified in the frequency domain, as observed in the field of neuroscience [5, 14, 6]. Concretely, for dynamical system components that belong to the same attractor, we apply a consistent evolution operator to derive their future states in the frequency domain. Our contributions can be summarized as follows:

- **A Chaos Lens on LTSF**. We incorporate chaos theory into LTSF tasks by leveraging the concept of attractor invariance, leading to a principal way to model the underlying continuous dynamics

- **Efficient Dynamic Modeling**. Our model Attraos employs a non-parametric embedding to obtain high-dimensional dynamical representations, leverages MDMU to capture the multi-scale dynamical structure, and performs the evolution in the frequency domain. Remarkably, Attraos achieves this with only about one-twelfth the parameter count of PatchTST. Furthermore, we utilize the Blelloch scan algorithm [3] to enable efficient computation of the MDMU.

- **Empirical Evidence**. Various experiments validate the superior performance of Attraos. Besides Leveraging the properties of chaotic dynamical systems, we explore their extended applications in LTSF tasks, focusing on chaotic evolution, modulation, representation, and reconstruction.

## 2 Preliminary

**Attractor in Chaos Theory**. In chaos theory, the interaction of three or more variables exhibiting periodic behavior gives rise to a complex dynamical system characterized by chaos. According to Takens's theorem [43, 34], assuming an ideal dynamical system $\mathcal{F} : \mathcal{M} \to \mathcal{M}$ that "lives" on attractor $\mathcal{A}$ in manifold space $\mathcal{M}$ which locally $\mathcal{C}^N$ (N-times differentiable), time series data $\{z_i\} \in \mathbb{R}$ can be interpreted as the observation of it by an unknown observation function $h$. To explore the properties of the unknown ideal dynamical system, we can employ the phase space reconstruction (PSR) method to establish an approximation $\mathcal{K} : \mathbb{R}^m \to \mathbb{R}^m$ which lives in differential homomorphism attractor $\tilde{\mathcal{A}}$ in the Euclidean space with suitable dimension $m$ [9]. The whole process is illustrated in Equation (1), where $\{z_i\}$, $\{u_i\}$ are the sampled data from two dynamical systems. Strictly speaking, in our paper, the chaotic attractor structure $\tilde{\mathcal{A}} = \{\tilde{\mathcal{A}}_i\}$ we focused on is in phase space $\mathbb{R}^m$. To facilitate understanding, we further provide a visual example of the Lorenz96 system in Appendix E.3.

In the forecasting stage, the local prediction method emerges as a prominent one: $u_{i+1} = \mathcal{K}^{(i)}(u_i)$, where the local evolution $\mathcal{K}^{(i)}$ can be either linear or nonlinear neural network [45, 2, 40], with the parameter being shared among the points in the neighborhood of $u_i$ or belong to the same local attractor. Considering the universal approximation capabilities of polynomials for dynamical systems, we leverage the polynomial to describe the chaotic dynamical structures.

$$
\begin{array}{ccc}
a_i \in \mathcal{A} \subset \mathcal{M} & \overset{\mathcal{F}}{\longmapsto} & a_{i+1} \in \mathcal{A} \subset \mathcal{M} \\
\downarrow h & & \downarrow h \\
z_i \in \mathbb{R} & & z_{i+1} \in \mathbb{R} \\
\downarrow PSR & & \downarrow PSR \\
u_i \in \tilde{\mathcal{A}} \subset \mathbb{R}^m & \overset{\mathcal{K}}{\longmapsto} & u_{i+1} \in \tilde{\mathcal{A}} \subset \mathbb{R}^m
\end{array} \quad (1)
$$

$$
x'(t) = \boldsymbol{A}x(t) + \boldsymbol{B}u(t) \quad (2\text{a})
$$
$$
x(t) = (K * u)(t) \quad (2\text{b})
$$
$$
K(t) = e^{t\boldsymbol{A}}\boldsymbol{B} \quad (2\text{c})
$$

**Polynomial Projection with Measure Window**. We only consider the first part of the SSM (2a), which is a parameterized map that transforms the input $u(t)$ into an $N$-dimensional latent space. According to Hippo [11], it is mathematically equivalent to: given an input $u(s)$, a set of orthogonal polynomial basis $\phi_n(t,s)$ that $\int_{-\infty}^{t} \phi_m(t,s)\phi_n(t,s)\mathrm{d}s = \delta_{m,n}$, and an inner product probability measure $\mu(t,s)$. This enables us to project the input $u(s)$ onto the polynomial basis along time dimension (3), and we can combine $\phi_n(t,s)\omega(t,s)$ as a kernel $K_n(t,s)$ (4). When $\omega(t,s)$ is defined in a time window $\mathbb{I}[t,t+\theta]$, it represents approximating the input over each window $\theta$.

$$
\langle u, \phi_n \rangle_\mu = \int_{-\infty}^{t} u(s)\phi_n(t,s)\omega(t,s)\mathrm{d}s \quad (3) \qquad x_n(t) = \int u(s)K_n(t,s)\mathbb{I}(t,s)\mathrm{d}s \quad (4)
$$

When the basis and measure are solely dependent on time $t$, it can be expressed in a convolution form (2b). In this paper, we will utilize this property to project the dynamical trajectories $\{u_n\}$ in phase space onto the polynomial spectral domain with kernel $e^{t\boldsymbol{A}}\boldsymbol{B}$ (2c) for characterization.

## 3 Theoretical Analysis & Methods

The overall structure of Attraos is illustrated in Figure 2. In this section, we provide a comprehensive description of its components, including the Phase Space Reconstruction embedding, the Multi-Resolution Dynamic Memory Unit (MDMU), and the frequency-enhanced local evolution, as well as the efficient computational methods employed.

### 3.1 Phase Space Reconstruction

According to chaos theory, the initial step involves constructing a topologically equivalent dynamical structure through the PSR. The preferred embedding method is typically the Coordinate Delay Reconstruction [34], which does not rely on any prior knowledge of the underlying dynamical system. By utilizing the discretely sampled data $\{z_i\}$ and incorporating two hyper-parameters, namely, embedding dimension $m$ and time delay $\tau$, a high-dimensional dynamical trajectory $\{u_i\}$ in phase space can be constructed by Eq. (5).

$$
u_i = (z_{i-(m-1)\tau}, z_{i-(m-2)\tau}, \cdots, z_i) \quad (5) \qquad \mathcal{K}_{patch} = \text{Unfold}\,(\mathcal{K}, p, p) \quad (6)
$$

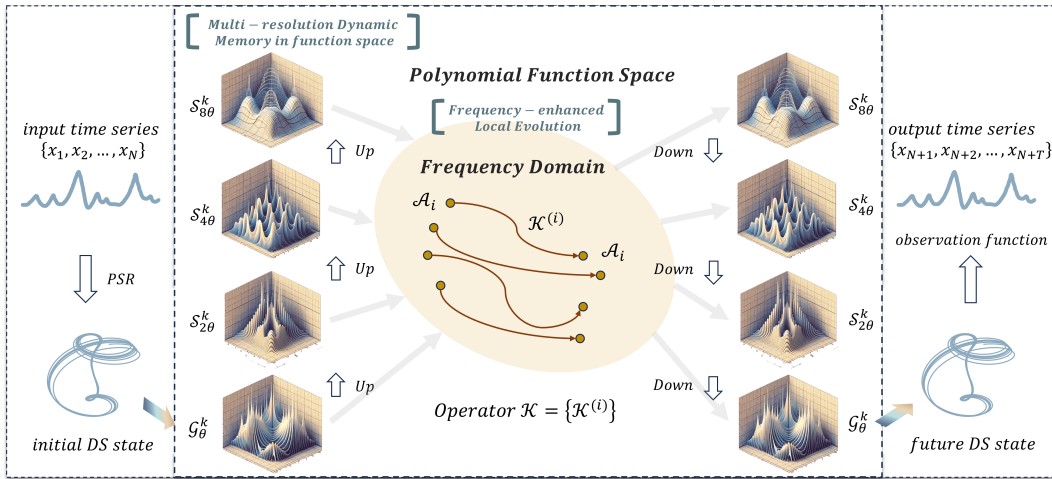

Figure 2: Overall architecture of Attraos. Initially, the PSR technique is employed to restore the underlying dynamical structures from historical data $\{z_i\}$. Subsequently, the dynamical system trajectory is fed into MDMU, projected onto polynomial space $\mathcal{G}_\theta^N$ using a time window $\theta$ and polynomial order $N$. Gradually, a hierarchical projection is performed to obtain more macroscopic memories of the dynamical system structure. Finally, local evolution operator $\mathcal{K}^{(i)}$ in the frequency domain is employed to obtain future state, thereby for the prediction.

For multivariate time series data with $C$ variables, we have observed considerable variations in the Lyapunov exponents of each variable, hence a channel-independent strategy [33] is employed to construct a unified dynamical system $\mathcal{K} \subset \mathbb{R}^m$. To accelerate model convergence and reduce complexity, we apply non-overlapping patching to obtain $\mathcal{K}_{patch}$ (6). We denote the number of patches as $L$, using $u \in \mathbb{R}^{B \times L \times D}$ to represent the tensor used for computing, where $D = mp$. The determination of $m$ and $\tau$ is achieved by applying the CC method [23] as shown in Appendix C.1.

**Remark 3.1.** *This represents the pioneering non-parametric embedding in LTSF, effectively reducing the model parameters in the embedding and output projection process ($m$ is typically single-digit).*

**Remark 3.2.** *In a large body of dynamical literature [44, 41, 20, 25], local linear approximation serves as an effective method for modeling dynamical systems, providing a basis for the effectiveness of the patching operations in this paper.*

## 3.2 Dynamical Representation by MDMU

**Proposition 1.** $A = \mathsf{diag}\{-1, -1, \dots\}$ *is a rough approximation of normal Hippo-LegT [11] matrix, which utilizes polynomial projection under a finite measure window (Lebesgue measure).*

**Remark 3.3.** *All proofs in this section can be found in Appendix B.*

Adhere to Mamba [10], we generate $B$ and measure window $\theta$ by linear layer, and propose a novel parameterized method (Proposition 1) for $A$ to instantiate Eq. (2a):

$$A = \mathsf{Broadcast}_D(\mathsf{diag}\{-1, -1, \dots\}), \qquad B = \mathsf{Linear}_B(u), \qquad \Delta/\theta = \mathsf{softplus}(\mathsf{Linear}_\Delta(u)), \tag{7}$$

where $A \in \mathbb{R}^{D \times N}$ represents the dynamical characteristics of the system's forward evolution in the polynomial space. Due to its diagonal nature, its representational capacity is comparable to $\mathbb{R}^{D \times N \times N}$; Matrix $B \in \mathbb{R}^{B \times L \times N}$ controls the process of projecting $u$ onto the polynomial domain like a gate mechanism; The learnable approximation window $\Delta \in \mathbb{R}^{B \times L \times D}$, similar to an attention mechanism, enables adaptive focus on specific dynamical structures (attractors).

**Remark 3.4.** *In the Hippo theory, the measure window is denoted by $\theta$, while in SSMs [10, 12], the discrete step size is represented by $\Delta$. These two terms can be considered approximately equivalent.*

As shown in Figure 3(a), in practical computations, we need to discretize Eq. (2a) to fit the discrete dynamical trajectories. We apply zero-order hold (ZOH) discretization [11] to matrix $A$, while opting for a combination of Forward Euler discretization for $B$ (instead of the commonly used

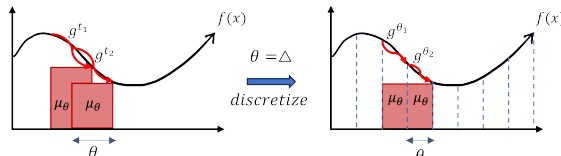

(a) Discretization for Limited Measure Approximation

(c) Sequential Computing $\mathcal{O}(L)$

(b) Multi-scale Dynamic Memory Unit

(d) Blelloch Scan Algorithm $\mathcal{O}(logL)$

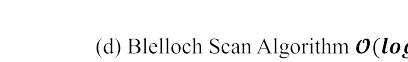

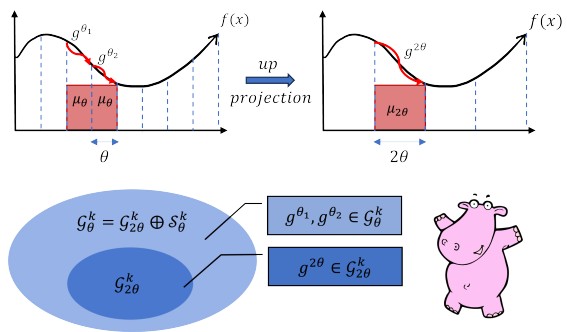

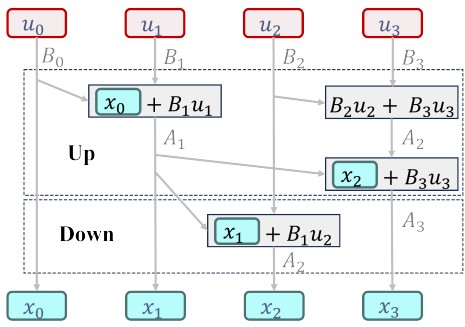

Figure 3: (a) Discretization of continuous polynomial approximation for sequence data. $g$ represents the optimal polynomial constructed from polynomial bases. (b) MDMU projects the dynamical structure onto different orthogonal subspaces $\mathcal{G}$ and $\mathcal{S}$. (c) Sequential computation for Eq. (2a) in $\mathcal{O}(L)$ time complexity. (d) Blelloch tree scanning for Eq. (2a) in $\mathcal{O}(logL)$ by storing intermediate results.

$\overline{B} = (\Delta A)^{-1}(\exp(\Delta A) - I) \cdot \Delta B$ in SSMs), resulting in a more concise representation:

$$(\textbf{ZOH}) : \overline{A} = \exp(\Delta A), \qquad (\textbf{Forward Euler}) : \overline{B} = \Delta B. \qquad (8)$$

Next, we can project the dynamical trajectory $u$ onto the polynomial domain using the discretized kernel to obtain dynamical representation $x \in \mathbb{R}^{B \times L \times D \times N}$ (9). This process can be achieved with $\mathcal{O}(L)$ complexity by employing sequential computation (Figure 3(c)) or by utilizing the Blelloch scan (Figure 3(d)) to store intermediate results with $\mathcal{O}(logL)$ complexity.

$$\overline{K} = \left( \overline{B}, \overline{AB}, \dots, \overline{A}^{L-1}\overline{B} \right), \qquad x = u * \overline{K}. \qquad (9)$$

Up to this point, the construction of the underlying continuous dynamical system $\mathcal{K}$ and its application to discretely sampled data $z_n$ have been established. However, in this scenario, we are still limited to a single representation of the dynamical structures with measure window $\theta$. The strange attractors, on the other hand, are often composed of multiple fundamental topological structures. Therefore, we require a multi-scale hierarchical representation of dynamical structures to capture their complexity.

To address this, as illustrated in Figure 3(b), we progressively increase the length of the window $\theta$ by powers of 2. The region previously approximated by $g^{\theta_1} \in \mathcal{G}_\theta^N$ (left half) and $g^{\theta_2} \in \mathcal{G}_\theta^N$ (right half) will now be approximated by $g^{2\theta} \in \mathcal{G}_{2\theta}^N$.

Since the piecewise polynomial function space can be defined as the following form:

$$\mathcal{G}_{(r)}^k = \begin{cases} g \mid \deg(g) < k, & x \in (2^{-r}l, 2^{-r}(l+1)) \\ 0, & \text{otherwise} \end{cases}, \qquad (10)$$

with polynomial order $k \in \mathbb{N}$, piecewise scale $r \in \mathbb{Z}^+ \cup \{0\}$, and piecewise internal index $l \in \{0, 1, ..., 2^r - 1\}$, it is evident that $dim(\mathcal{G}_{(r)}^k) = 2^r k$, implying that $\mathcal{G}_\theta^k$ possesses a superior function capacity compared to $\mathcal{G}_{2\theta}^k$. All functions in $\mathcal{G}_{2\theta}^k$ are encompassed within the domain of $\mathcal{G}_\theta^k$. Moreover, since $\mathcal{G}_\theta^k$ and $\mathcal{G}_{2\theta}^k$ can be represented as space spanned by basis functions $\{\phi_i^\theta(x)\}$ and $\{\phi_i^{2\theta}(x)\}$, any function including the basis function within the $\mathcal{G}_{2\theta}^k$ space can be precisely expressed as a linear combination of basis functions from the $\mathcal{G}_\theta^k$ space with a proper tilted measure $\mu_{2\theta}$:

$$\phi_i^{2\theta}(x) = \sum_{j=0}^{k-1} H_{ij}^{\theta_1} \phi_i^\theta(x)_{x \in [\theta_1]} + \sum_{j=0}^{k-1} H_{ij}^{\theta_2} \phi_i^\theta(x)_{x \in [\theta_2]}, \qquad (11)$$

The projection coefficients on these two spaces can be mutually transformed using the linear projection matrix $H$ and its inverse matrix $H^\dagger$ based on the odd or even positions along the $L$ dimension in $x$.

$$x^{2\theta} = H^{\theta_1} x^{\theta_1} + H^{\theta_2} x^{\theta_2}, \qquad x^{2\theta} \in \mathbb{R}^{B \times L/2 \times D \times N} \tag{12}$$

**Remark 3.5.** *Although $\Delta$ is akin to an attention mechanism leads to different measure windows, ie., $\theta_1 \neq \theta_2$, it still maintains the linear projection property for up and down projection:$\mathcal{G}_{\theta_1}, \mathcal{G}_{\theta_2} \leftrightarrow \mathcal{G}_{\theta_1 + \theta_2}$. In our illustration, we have used a unified measure window for simplicity.*

**Theorem 2.** *(Approximation Error Bound) Function $f : [0,1] \in \mathbb{R}$ is $k$ times continuously differentiable, the piecewise polynomial $g \in \mathcal{G}_r^N$ approximates $f$ with mean error bounded as:*

$$\|f - g\| \leq 2^{-rN} \frac{2}{4^N N!} \sup_{x \in [0,1]} \left| f^{(N)}(x) \right|.$$

Iteratively repeating this process enables us to model the dynamical structure from a more macroscopic perspective. Theorem 2 indicates that under this approach, the polynomial projection error has convergence of order $N$. Hyper-parameter analysis can be found in Figure 5.

**Remark 3.6.** *When the weight function is uniformly equal across the dynamical structure, $H^{\theta_1}$ and $H^{\theta_2}$ are shared in each projection level as the projection matrix for the left and right interval.*

**Theorem 3.** *The mean **attractor evolution error** $\left\| \mathcal{K} \circ \tilde{\mathcal{A}} - \tilde{\mathcal{A}} \right\|$ of evolution operator $\mathcal{K} = \{\mathcal{K}^{(i)}\}$ is bounded by $\|\mathcal{K} \circ \tilde{\mathcal{A}} - \tilde{\mathcal{A}}\|(N-1)\mathcal{E}\left(-\beta \nabla_i + 1\right)$, with the number of random patterns $N \geq \sqrt{p} c \frac{d-1}{4}$ stored in the system by interaction paradigm $\mathcal{E}$ in an ideal spherical retrieval paradigm.*

Based on this hierarchical projection, we additionally want to uphold the constancy of attractor patterns throughout the evolution, which is equivalent to minimizing attractor evolution errors. According to Theorem 3, it is imperative to ensure the separation between attractors, denoted as $\nabla_i := \min_{j, j \neq i} \left( \tilde{\mathcal{A}}_i^T \tilde{\mathcal{A}}_i - \tilde{\mathcal{A}}_i^T \tilde{\mathcal{A}}_j \right)$, is sufficiently large. While there is an intersection between the $\mathcal{G}_\theta^N$ and $\mathcal{G}_{2\theta}^N$ spaces, which limits the attainment of a sufficiently large $\nabla$. To address this issue, we define an orthogonal complement space as $\mathcal{G}_\theta^N = \mathcal{S}_\theta^N \bigoplus \mathcal{G}_{2\theta}^N$, to establish a series of orthogonal function spaces $\{\mathcal{S}_\theta^N, \mathcal{S}_{2\theta}^N, ..., \mathcal{S}_{2^L \theta}^N, \mathcal{G}_{2^L \theta}^N\}$. We can extend Eq. (12) as:

$$x^{2\theta} = H^{\theta_1} x^{\theta_1} + H^{\theta_2} x^{\theta_2}, \quad s^{2\theta} = G^{\theta_1} x^{\theta_1} + G^{\theta_2} x^{\theta_2}, \tag{13}$$

$$x^{\theta_1} = H^{\dagger \theta_1} x^{2\theta} + G^{\dagger \theta_1} s_t^{2\theta}, \quad x^{\theta_2} = H^{\dagger \theta_2} x^{2\theta} + G^{\dagger \theta_2} s_t^{2\theta}. \tag{14}$$

Theorem 4 states that the coarsest-grained $\mathcal{G}$ space, along with a series of orthogonal complement $\mathcal{S}$ spaces, can approximate any given dynamical system structure with finite error bounds in Theorem 2.

**Theorem 4.** *(Completeness in $L^2$ space) The orthonormal system $B_N = \{\phi_j : j = 1, \ldots, N\} \cup \{\psi_j^{rl} : j = 1, \ldots, N; r = 0, 1, 2, \ldots; l = 0, \ldots, 2^r - 1\}$ spans $L^2[0, 1]$.*

**Remark 3.7.** *$\boldsymbol{H}, \boldsymbol{G}, \boldsymbol{H}^\dagger, \boldsymbol{G}^\dagger \in \mathbb{R}^{N \times N}$ are obtained by applying Gaussian Quadrature to Legendre polynomials [13]. The gradients of these matrices are subsequently utilized for adaptive optimization.*

The hierarchical projection can be implemented explicitly using iterative display or can be efficiently computed through an implicit implementation proposed in the following Section 3.3.

## 3.3 System Evolution by Attractors

Following chaos theory, we employ a local evolution method $x_{i+1} = \mathcal{K}^{(i)}(x_n)$ to forecast future state. Given dynamics representations $s/x$, the subsequent step involves partitioning the attractor and utilizing the operator $\mathcal{K}^i$ belonging to $A_i$ for system evolution. We present three evolution strategies:

- *Direct Evolution:* We employ each representation $x_t/s_t \in \mathbb{R}^{D \times N}$ as the feature to partition adjacent points in the dynamical system trajectory into the same attractor using the K-means method. Subsequently, we evolve these points using a local operator $\mathcal{K}^{(i)}$.

- *Frequency-enhanced Evolution:* Inspired by neuroscience [6], where attractor structures are amplified in the frequency domain, we first obtain the frequency domain representation of the dynamical structure through Fourier transformation. Considering the dominant modes as attractors, we employ $\mathcal{K}^{(i)}$ to drive the system's evolution in the frequency domain.

- ***Hopfield Evolution:*** Hopfield networks [16] are designed specifically for attractor memory retrieval (see Appendix A.2). In our approach, we utilize a modern version [38] of the Hopfield network for the evolution of dynamical systems, employing cross-attention operations. We treat the trainable attractor library as the *Key* and *Value*, while different scales of dynamical structure representations $\{s^\theta, s^{2\theta}, ..., s^{2^L\theta}, x^{2^L\theta}\}$ serve as the *Query*, enabling sequence-to-sequence evolution.

Our experimental results in Table 4 demonstrate that the frequency-enhanced evolution strategy outperforms others comprehensively, and we introduce two implementation approaches:

**Explicit Evolution**. Initially, the finest dynamical representation $x^\theta$ is obtained using $\overline{K}$, and then expanded to multiple scales $s^\theta, s^{2\theta}, ..., s^{2^L\theta}, x^{2^L\theta}$. By applying the Fourier Transform, we select $M$ low-frequency components as the primary modes. Each mode undergoes linear evolution $\mathcal{K}^{(i)} = W_i \in \mathbb{C}^{N \times N}$, followed by back projection to the original scale.

**Implicit Evolution**. However, explicit evolution methods inevitably increase the time complexity. To address this issue, inspired by the Blelloch algorithm and hierarchical projection, which both utilize ***tree-like computational graphs***, we propose an implicit evolution method.

A Blelloch scan [3, 42] (Figure 3(d)) defines a binary operator $q_i \bullet q_j := (q_{j,a} \odot q_{i,a}, \ q_{j,a} \otimes q_{i,b} + q_{j,b})$ used to compute the linear recurrence $x_k = \overline{A}x_{k-1} + \overline{B}u_k$. We take $L = 4$ as an example:



**Up sweep for even position**       **Down sweep for odd position**



$$r_2 = c_1 \bullet c_2 = \left(\overline{A}, \overline{B}u_1\right) \bullet \left(\overline{A}, \overline{B}u_2\right) = \left(\overline{A}^2, \overline{AB}u_1 + \overline{B}u_2\right) \quad r_1 = r_0 \bullet c_1 = (I, 0) \bullet \left(\overline{A}, \overline{B}u_1\right) = \left(\overline{A}, \overline{B}u_1\right)$$

$$q_4 = c_3 \bullet c_4 = \left(\overline{A}, \overline{B}u_3\right) \bullet \left(\overline{A}, \overline{B}u_4\right) = \left(\overline{A}^2, \overline{AB}u_3 + \overline{B}u_4\right) \quad r_3 = r_2 \bullet c_3 = \left(\overline{A}^2, \overline{AB}u_1 + \overline{B}u_2\right) \bullet \left(\overline{A}, \overline{B}u_3\right)$$

$$r_4 = r_2 \bullet q_4 = \left(\overline{A}^2, \overline{AB}u_1 + \overline{B}u_2\right) \bullet \left(\overline{A}^2, \overline{AB}u_3 + \overline{B}u_4\right) \qquad\qquad = \left(\overline{A}^3, \overline{A}^2\overline{B}u_1 + \overline{AB}u_2 + \overline{B}u_3\right)$$

$$= \left(\overline{A}^4, \overline{A}^3\overline{B}u_1 + \overline{A}^2\overline{B}u_2 + \overline{AB}u_3 + \overline{B}u_4\right). \qquad x_1, x_2, x_3, x_4 = r_1[R], r_2[R], r_3[R], r_4[R].$$

The process commences by computing the values of variable $x$ at even positions through an upward sweep. Subsequently, these even position values are employed during a downward sweep to calculate the values at odd positions. The corresponding value of $x$ resides in the right node of $r$. Thus, we can modify the binary operator as $q_i \bullet q_j := (q_{j,a} \odot q_{i,a}, \ H_i \otimes (q_{j,a} \otimes q_{i,b} + q_{j,b}))$, thereby implicitly integrating hierarchical projection into the scanning operation. This leads to the scales of $x_i$:

$$scale(x_i) = \begin{cases} 0 & \text{if } i = 0 \\ scale(x_{i-1}) + 1 & \text{if } i \text{ is odd} \\ \log_2(i) & \text{if } i \text{ is even and a power of 2} \\ 1 & \text{if } i \text{ is even and not a power of 2} \end{cases}$$

We simplify the original $H^{\theta_1/\theta_2}, G^{\theta_1/\theta_2}, H^{\dagger\theta_1/\theta_2}, G^{\dagger\theta_1/\theta_2}$ with just $H \in \mathbb{R}^{B \times L \times N}$, which is generated directly through a linear layer, and omit the reconstruction process. This approach directly sparsifies the kernels $e^{\overline{A}t}B$ in different subspaces and using the linear layer to generate hierarchical space projection matrix. Afterwards, we learn the evolution using the data $x$ in the frequency domain.

$$H = \mathsf{Linear}_{H}(u), \qquad W_{out} = \mathsf{Linear}_{W_{out}}(u). \tag{15}$$

In this paper, we utilize this indirect and efficient hierarchical projection as the default setting. Ultimately, Attraos projects from the polynomial spectral space back to the phase space by employing another gating projection $W_{out} \in \mathbb{R}^{B \times L \times N \times 1}$ (15) to $x \in \mathbb{R}^{B \times L \times D \times N}$, and derives the prediction results using an observation function parameterized by $W_h \in \mathbb{R}^{LD \times H}$ (flattening the patches).

## 4 Experiments

In this section, we commence by conducting a comprehensive performance comparison of Attraos against other state-of-the-art models in seven mainstream LTSF datasets along with two typical chaotic datasets, namely Lorenz96-3d and Air-Convection, followed by ablation experiments pertaining to model architectures. Furthermore, leveraging the properties of chaotic dynamical systems, we explore their extended applications, including experiments on chaotic evolution, representation, reconstruction, and modulation (Appendix E). Finally, we provide the complexity analysis and robustness analysis. For detailed information regarding baseline models, dataset descriptions, experimental settings, and hyper-parameter analysis, please refer to Appendix D.

Table 1: Average results of long-term forecasting with an input length of 96 and prediction horizons of {96, 192, 336, 720}. The best performance is in Red, and the second best is in Blue. Full results are in Appendix E.5.

| Model | Attraos (Ours) | | Mamba4TS (Time Emb.) | | S-Mamba [47] | | RWKV-TS [17] | | GPT-TS [55] | | Koopa [32] | | InvTrm [31] | | PatchTST [33] | | DLinear [52] | |
|---|---|---|---|---|---|---|---|---|---|---|---|---|---|---|---|---|---|---|
| Metric | MSE | MAE | MSE | MAE | MSE | MAE | MSE | MAE | MSE | MAE | MSE | MAE | MSE | MAE | MSE | MAE | MSE | MAE |
| ETTh1 | 0.423 | 0.420 | 0.444 | 0.438 | 0.459 | 0.453 | 0.454 | 0.446 | 0.457 | 0.450 | 0.450 | 0.443 | 0.463 | 0.454 | 0.434 | 0.435 | 0.462 | 0.458 |
| ETTh2 | 0.372 | 0.399 | 0.386 | 0.410 | 0.381 | 0.407 | 0.375 | 0.402 | 0.389 | 0.414 | 0.397 | 0.417 | 0.383 | 0.407 | 0.380 | 0.406 | 0.564 | 0.520 |
| ETTm1 | 0.382 | 0.391 | 0.396 | 0.406 | 0.399 | 0.407 | 0.391 | 0.403 | 0.396 | 0.401 | 0.395 | 0.403 | 0.407 | 0.412 | 0.403 | 0.398 | 0.403 | 0.406 |
| ETTm2 | 0.280 | 0.324 | 0.299 | 0.343 | 0.289 | 0.333 | 0.285 | 0.330 | 0.294 | 0.339 | 0.281 | 0.326 | 0.291 | 0.335 | 0.283 | 0.329 | 0.345 | 0.396 |
| Exchange | 0.349 | 0.395 | 0.364 | 0.405 | 0.364 | 0.407 | 0.406 | 0.439 | 0.371 | 0.409 | 0.390 | 0.424 | 0.366 | 0.416 | 0.383 | 0.416 | 0.346 | 0.416 |
| Crypto | 0.187 | 0.157 | 0.193 | 0.162 | 0.198 | 0.163 | 0.190 | 0.159 | 0.196 | 0.164 | 0.199 | 0.165 | 0.196 | 0.164 | 0.192 | 0.161 | 0.201 | 0.176 |
| Weather | 0.246 | 0.271 | 0.258 | 0.280 | 0.252 | 0.277 | 0.256 | 0.280 | 0.279 | 0.279 | 0.247 | 0.273 | 0.260 | 0.280 | 0.258 | 0.280 | 0.267 | 0.319 |

Table 2: Prediction results on the artificial (Lorenz96-3d) and real-world chaotic datasets (Air-convection) with various forecasting lengths. Red/Blue denotes the best/second performance.

| | Model | Attraos (Ours) | | Mamba4TS (Time Emb.) | | S-Mamba [47] | | RWKV-TS [17] | | Koopa [32] | | InvTrm [31] | | PatchTST [33] | | DLinear [52] | |
|---|---|---|---|---|---|---|---|---|---|---|---|---|---|---|---|---|---|
| | Metric | MSE | MAE | MSE | MAE | MSE | MAE | MSE | MAE | MSE | MAE | MSE | MAE | MSE | MAE | MSE | MAE |
| Lorenz96-3d | 96 | 0.844 | 0.684 | 0.892 | 0.721 | 0.925 | 0.744 | 0.894 | 0.722 | 0.891 | 0.736 | 0.963 | 0.786 | 0.929 | 0.756 | 0.881 | 0.750 |
| | 192 | 0.835 | 0.662 | 0.910 | 0.748 | 0.917 | 0.761 | 0.894 | 0.744 | 0.881 | 0.752 | 0.944 | 0.811 | 0.899 | 0.714 | 0.910 | 0.753 |
| | 336 | 0.837 | 0.681 | 0.943 | 0.772 | 0.968 | 0.788 | 0.982 | 0.823 | 0.914 | 0.753 | 0.997 | 0.841 | 0.922 | 0.787 | 0.893 | 0.737 |
| | 720 | 0.872 | 0.739 | 0.996 | 0.814 | 1.135 | 0.940 | 1.058 | 0.921 | 0.989 | 0.801 | 1.129 | 0.955 | 0.971 | 0.828 | 0.927 | 0.806 |
| | AVG | 0.847 | 0.692 | 0.935 | 0.764 | 0.986 | 0.808 | 0.957 | 0.803 | 0.919 | 0.761 | 1.008 | 0.848 | 0.930 | 0.771 | 0.903 | 0.762 |
| Air | 96 | 0.437 | 0.303 | 0.451 | 0.314 | 0.468 | 0.329 | 0.447 | 0.308 | 0.443 | 0.307 | 0.470 | 0.337 | 0.465 | 0.331 | 0.441 | 0.325 |
| | 192 | 0.455 | 0.321 | 0.472 | 0.331 | 0.481 | 0.340 | 0.467 | 0.328 | 0.451 | 0.329 | 0.485 | 0.349 | 0.477 | 0.341 | 0.460 | 0.338 |
| | 336 | 0.456 | 0.334 | 0.468 | 0.342 | 0.485 | 0.351 | 0.461 | 0.339 | 0.468 | 0.342 | 0.499 | 0.363 | 0.484 | 0.353 | 0.461 | 0.341 |
| | 720 | 0.466 | 0.355 | 0.492 | 0.379 | 0.501 | 0.386 | 0.482 | 0.367 | 0.488 | 0.369 | 0.516 | 0.401 | 0.504 | 0.392 | 0.474 | 0.359 |
| | AVG | 0.454 | 0.328 | 0.471 | 0.342 | 0.484 | 0.352 | 0.464 | 0.336 | 0.463 | 0.337 | 0.493 | 0.363 | 0.483 | 0.354 | 0.459 | 0.341 |

## 4.1 Overall Performance

**Mainstream LTSF Datasets**. As depicted in Table 1, we can observe that: (a) Attraos consistently exhibits the best performance, closely followed by RWKV-TS and Koopa, which underscores the crucial role of modeling temporal dynamics in LTSF tasks. (b) The models based on state space models (Mamba4TS, S-Mamba) generally outperform the Transformer-based models (PatchTST, InvTrm), indicating the potential superiority of state space models as fundamental frameworks for temporal modeling. (c) The performance of the GPT-TS model, which relies on a pre-trained large language model, is relatively average, suggesting the inherent challenge in directly capturing the dynamics of temporal data using such models. A promising avenue for future research lies in training a dynamical foundational model from scratch on large-scale physical datasets or leveraging pre-training to obtain an attractor tokenizer that is better suited for inputs to the large language model.

**Chaotic Datasets**. Table 2 presents the results on both artificial and real-world chaotic datasets. It can be observed that: (a) Attraos exhibits superior performance on both datasets, thanks to the utilization of the PSR and MDMU modules, which effectively capture the multi-scale attractor structures. (b) In Lorenz96 dataset, where there is a prior knowledge about the phase space dimension, Attraos outperforms other models by a significant margin. This highlights the importance of PSR in recovering the complete temporal dynamics. (c) Apart from Attraos, the linear model (DLinear) demonstrate the best predictive results due to their robustness. Conversely, deep learning models based on transformers exhibit weaker performance and fail to model chaotic dynamics effectively.

## 4.2 Further Analysis

**Ablation Studies.** Next, we turn off each module of Attraos to assess their individual effects. As shown in Table 3, we observe consistent performance decline of Attraos when deleting each module: (a) The removal of Phase Space Reconstruction exhibits the most severe performance degradation, indicating that the process of reconstructing the dynamical structure through PSR forms the foundation for Attraos' efficient capture of attractor structures. (b) Multi-scale hierarchical projection effectively captures the complex topological structure of the singular attractor, leading to improved performance. However, overfitting may occur in certain prediction lengths. (c) Time-varying $B$ and $W_{out}$ acts as a gating attention mechanism, allowing for a more focused emphasis on dynamical structure segments

Table 3: Results of ablation study. "w/o" denotes without. PSR: Phase Space Reconstruction; MS: Multi-scale hierarchical projection; TV: Time-varying $B$ and $W_{out}$; SPA: Specially initialized $A$ ; FE: Frequency Evolution. Red/Blue denotes the performance improvement/decline.

| Model | | Attraos | | w/o PSR | | w/o MS | | w/o TV | | w/o SPA | | w/o FE | |
|---|---|---|---|---|---|---|---|---|---|---|---|---|---|
| Metric | | MSE | MAE | MSE | MAE | MSE | MAE | MSE | MAE | MSE | MAE | MSE | MAE |
| ETTh2 | 96 | 0.292 | 0.348 | 0.301 | 0.357 | 0.299 | 0.353 | 0.299 | 0.354 | 0.294 | 0.351 | 0.297 | 0.352 |
| | 192 | 0.374 | 0.386 | 0.389 | 0.405 | 0.384 | 0.393 | 0.381 | 0.395 | 0.373 | 0.384 | 0.378 | 0.388 |
| | 336 | 0.420 | 0.432 | 0.427 | 0.438 | 0.426 | 0.436 | 0.424 | 0.430 | 0.425 | 0.436 | 0.427 | 0.435 |
| | 720 | 0.418 | 0.431 | 0.431 | 0.450 | 0.425 | 0.437 | 0.416 | 0.427 | 0.421 | 0.433 | 0.427 | 0.437 |
| | AVG | 0.376 | 0.399 | 0.387 | 0.413 | 0.380 | 0.405 | 0.380 | 0.402 | 0.478 | 0.401 | 0.382 | 0.403 |
| Weather | 96 | 0.159 | 0.206 | 0.171 | 0.215 | 0.163 | 0.210 | 0.167 | 0.214 | 0.162 | 0.209 | 0.164 | 0.211 |
| | 192 | 0.212 | 0.249 | 0.266 | 0.263 | 0.218 | 0.253 | 0.222 | 0.270 | 0.215 | 0.249 | 0.216 | 0.253 |
| | 336 | 0.265 | 0.288 | 0.282 | 0.304 | 0.271 | 0.295 | 0.277 | 0.297 | 0.263 | 0.288 | 0.270 | 0.294 |
| | 720 | 0.347 | 0.340 | 0.358 | 0.351 | 0.346 | 0.338 | 0.355 | 0.346 | 0.347 | 0.342 | 0.355 | 0.356 |
| | AVG | 0.246 | 0.271 | 0.259 | 0.283 | 0.250 | 0.274 | 0.255 | 0.282 | 0.247 | 0.272 | 0.251 | 0.279 |

Table 4: Results of various chaotic evolution strategies. Red/Blue denotes the best/second performance.

| SSMs | | Implicit Fre | | Explicit Fre | | Direct-Linear | | Direct-CNN | | Hopfield-16modes | | Hopfield-64modes | |
|---|---|---|---|---|---|---|---|---|---|---|---|---|---|
| Metric | | MSE | MAE | MSE | MAE | MSE | MAE | MSE | MAE | MSE | MAE | MSE | MAE |
| ETTh1 | 96 | 0.370 | 0.388 | 0.376 | 0.392 | 0.388 | 0.404 | 0.395 | 0.410 | 0.384 | 0.405 | 0.389 | 0.407 |
| | 192 | 0.416 | 0.418 | 0.419 | 0.423 | 0.441 | 0.439 | 0.444 | 0.437 | 0.430 | 0.446 | 0.427 | 0.442 |
| | 336 | 0.458 | 0.432 | 0.465 | 0.439 | 0.488 | 0.460 | 0.482 | 0.456 | 0.480 | 0.482 | 0.485 | 0.489 |
| | 720 | 0.447 | 0.442 | 0.454 | 0.448 | 0.511 | 0.508 | 0.510 | 0.512 | 0.494 | 0.491 | 0.502 | 0.500 |
| | AVG | 0.423 | 0.420 | 0.429 | 0.426 | 0.457 | 0.453 | 0.458 | 0.454 | 0.447 | 0.456 | 0.426 | 0.460 |
| ETTm2 | 96 | 0.172 | 0.254 | 0.175 | 0.258 | 0.187 | 0.266 | 0.191 | 0.269 | 0.181 | 0.260 | 0.182 | 0.263 |
| | 192 | 0.242 | 0.301 | 0.247 | 0.308 | 0.264 | 0.331 | 0.265 | 0.334 | 0.255 | 0.312 | 0.259 | 0.314 |
| | 336 | 0.303 | 0.340 | 0.310 | 0.349 | 0.325 | 0.359 | 0.319 | 0.354 | 0.315 | 0.347 | 0.312 | 0.344 |
| | 720 | 0.401 | 0.399 | 0.407 | 0.405 | 0.424 | 0.419 | 0.421 | 0.422 | 0.420 | 0.426 | 0.417 | 0.422 |
| | AVG | 0.280 | 0.324 | 0.285 | 0.330 | 0.300 | 0.344 | 0.297 | 0.345 | 0.293 | 0.336 | 0.293 | 0.336 |

that potentially contain attractor structures, thereby enhancing performance. (d) The initialization method proposed for $A$ matrix demonstrates marginal yet consistent improvements, underscoring the importance of prior inductive bias for machine learning models. (e) Frequency domain evolution methods significantly reduce temporal noise information and amplify attractor structures. We will further analyze the importance of frequency domain evolution in subsequent analysis.

**Chaotic Evolution Strategy.** We further compare various dynamical system evolution strategies mentioned in Section 3.3. From Table 4, it is evident that the frequency-enhanced evolution strategy outperforms the others. Moreover, our proposed efficient implicit evolution method can adaptively explore multi-scale dynamical structure information, avoiding redundant cyclic computations and mitigating overfitting. (b) An inherent characteristic of time series data is significant noise, making it challenging to capture the underlying dynamical structures in the time domain. Direct evolution strategies, whether linear or non-linear neural network-based, do not yield satisfactory results. Moreover, according to the theorem 2 in FiLM [53], the recursion computation for dynamic projection further accumulates noise information. (c) Applying Hopfield networks in the time domain also proves to be unsatisfactory, and even adding more patterns (*Key* and *Value*) can have adverse effects. A potential solution is to apply Hopfield networks in the frequency domain instead.

**Complexity Analysis.** As depicted in Figure 4, we present a comprehensive visual analysis comparing Attraos with various baseline models in terms of their average performance on the ETTh1 dataset. The x-axis represents the training time, the y-axis represents the test loss, and the circle radius corresponds to the model parameters. In this analysis, we substituted GPT-TS with FiLM due to its limited relevance to this specific evaluation. The results clearly demonstrate that Attraos surpasses other models in both time and space complexity, maintaining a significant advantage. Notably, when compared to the PatchTST model with a hidden dimension of 256 (2.4M parameters), Attraos (0.2M parameters) possesses only one-twelfth of its parameter count.

**Robustness Analysis.** We add a 0.1 * $\mathcal{N}(0, 1)$ Gaussian noise to the training dataset to test the robustness of Attraos. As shown in Table 5, it can be observed that Attraos exhibits strong robustness against noisy data, and increasing the level of noise can even lead to further performance improvement. This is attributed to the frequency domain evolution strategy, where we retain only the dominant modes as attractor structures, effectively removing the noise information. Furthermore, an interesting phenomenon has been observed in our experiments: as noise is introduced, the model's convergence speed increases. This discovery warrants further exploration in future studies.

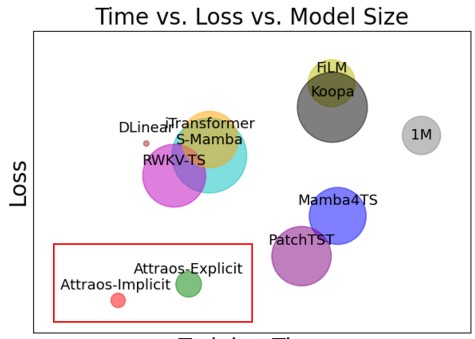

Figure 4: Complexity analysis.

| Model | | Attraos | | Attrao-noise | |
|---|---|---|---|---|---|
| Metric | | MSE | MAE | MSE | MAE |
| ETTh1 | 96 | 0.370 | 0.388 | **0.360** | **0.390** |
| | 192 | 0.416 | 0.418 | **0.413** | **0.415** |
| | 336 | 0.458 | 0.432 | **0.455** | **0.430** |
| | 720 | 0.447 | 0.442 | **0.451** | **0.444** |
| | AVG | 0.423 | 0.420 | **0.422** | 0.420 |
| ETTm2 | 96 | 0.172 | 0.254 | **0.170** | **0.251** |
| | 192 | 0.242 | 0.301 | **0.238** | **0.297** |
| | 336 | 0.303 | 0.340 | **0.305** | **0.339** |
| | 720 | 0.401 | 0.399 | **0.398** | **0.392** |
| | AVG | 0.280 | 0.324 | **0.278** | **0.320** |

Table 5: Robustness analysis with additional noise. Red/Blue denotes the performance improvement/decline.

**Chaotic Reconstruction.** As illustrated in the left of Figure 5, we visualize the phase space forecasting results of Attraos on the Lorenz96 system. It can be observed that: Although some minor details may be missing due to the sparsity introduced by the frequency domain evolution, Attraos successfully reconstructs the chaotic dynamic structure of Lorenz96. Moreover, modeling time series based on the dynamics structure of the phase space can be viewed as a form of data augmentation, e.g., two-dimensional time figuring [50] or seasonal decomposition [54].

**Chaotic Representation & Hyper-parameter Analysis.** In the right of Figure 5, we validated the impact of polynomial dimensions on the model performance. Noteworthy observations include: As noted in substantial literature on SSMs, a polynomial dimension of 256 is generally required to approximate input time series signals with sufficient accuracy. However, we found that the patch operation effectively reduces this threshold in a linear fashion. For instance, in the ETT dataset with a phase space dimension of 4, the required polynomial dimension drops to 256/4 = 64 dimensions.

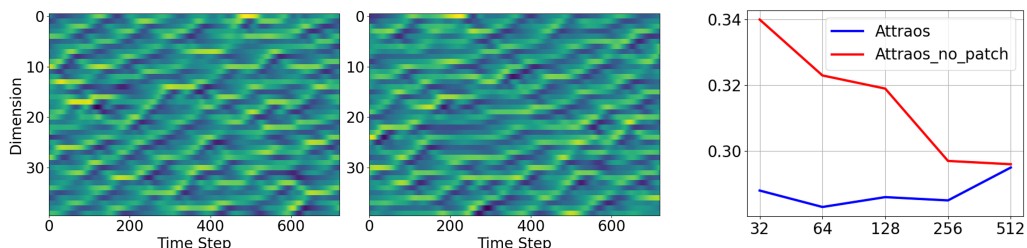

Figure 5: Left: Chaotic Reconstruction for Lorenz96 system with 720 forecasting step. Right: Hyper-parameter analysis w.r.t. polynomial orders for different model variants w.r.t. patching operation in ETTm2 dataset.

**Discussion.** Recently, across various fields of machine learning, an increasing number of works have focused on the underlying physical properties hidden within real-world observational data [20, 21, 26, 51]. By leveraging physical priors, these models achieve significant improvements in generalization, accuracy, and interpretability. We hope Attraos introduces a fresh perspective to the LTSF community and encourages the emergence of physics-guided time series analysis models.

## 5 Conclusion and Future Work

LTSF tasks have long been a focal point of research in the machine-learning community. However, mainstream deep learning models currently overlook the crucial aspect that time series data is derived from discretely sampling underlying continuous dynamical systems. Inspired by chaotic theory, our model, Attraos, considers time series as generated by a generalized chaotic dynamical system. By leveraging the invariance of attractors, Attraos enhances predictive performance and provides empirical and theoretical explanations. In the future, we will verify our proposed Attraos on large-scale chaotic datasets and utilize the implicit neural representations for the phase space coordinates, enabling a trainable embedding process to address the instability of PSR techniques.

## Acknowledgments and Disclosure of Funding

This work is mainly supported by the National Natural Science Foundation of China (No. 62402414). This work is also supported by the Guangzhou-HKUST(GZ) Joint Funding Program (No. 2024A03J0620), Guangzhou Municipal Science and Technology Project (No. 2023A03J0011), the Guangzhou Industrial Information and Intelligent Key Laboratory Project (No. 2024A03J0628), and a grant from State Key Laboratory of Resources and Environmental Information System, and Guangdong Provincial Key Lab of Integrated Communication, Sensing and Computation for Ubiquitous Internet of Things (No. 2023B1212010007).

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

# A Technical Background

## A.1 Takens Theorem

Takens' theorem, introduced by Floris Takens in 1981, provides a framework for reconstructing the dynamics of a chaotic system based on a series of observations. The theorem establishes conditions under which the state space of a dynamical system can be reconstructed from time-delay measurements of a single observable. The reconstructed system retains properties that are invariant under smooth transformations, known as diffeomorphisms.

In the context of discrete-time dynamical systems, Takens' theorem is commonly applied. Consider a dynamical system whose state space is a $v$-dimensional manifold $\mathcal{M}$, with the evolution of the system governed by a smooth map

$$\mathcal{F} : \mathcal{M} \to \mathcal{M}.$$

Assume there exists a strange attractor $\mathcal{A} \subset \mathcal{M}$ with a box-counting dimension $d_A$. According to Whitney's embedding theorem [37], the attractor $\mathcal{A}$ can be embedded into an $m$-dimensional Euclidean space if

$$m > 2d_A.$$

This implies that there is a diffeomorphism $\varphi$ mapping the attractor $\mathcal{A}$ into $\mathbb{R}^N$, such that the Jacobian matrix of $\varphi$ is of full rank. In the delay embedding process, the embedding function is constructed using an observation function. This observation function $h : \mathcal{M} \to \mathbb{R}$ needs to be twice continuously differentiable and must assign a real number to each point on the attractor $\mathcal{A}$, ensuring that it is typical, meaning its derivative is of full rank without exhibiting any special symmetries.

The delay embedding theorem states that the mapping

$$\varphi_T(x) = \left( h(x), h(\mathcal{F}(x)), \ldots, h\left(\mathcal{F}^{k-1}(x)\right) \right)$$

constitutes an embedding of the strange attractor $\mathcal{A}$ into $\mathbb{R}^N$.

Now, consider a $d$-dimensional state vector $x_t$, which evolves according to an unknown but deterministic and continuous dynamic process. Suppose a one-dimensional observable $y$ exists, which is a smooth function of $x$ and is coupled to all the components of $x$. At any given time, we can observe not only the current measurement $y(t)$ but also measurements from past times separated by a lag $\tau$: $y_{t+\tau}, y_{t+2\tau}$, and so on. Using $m$ such lags results in an $m$-dimensional vector. As the number of lags increases, the motion in the reconstructed space becomes more predictable, and in the limit as $m \to \infty$, the dynamics could become deterministic. In reality, the deterministic nature of the dynamics is achieved at a finite dimension, with the reconstructed dynamics being equivalent to the original system's dynamics, related by a smooth, invertible change of coordinates (a diffeomorphism). The theorem specifically asserts that deterministic behavior emerges once the dimension reaches $2d + 1$, with the minimal embedding dimension often being lower.

## A.2 Hopfield Network

### A.2.1 Classical Hopfield Network

A Hopfield network is a form of recurrent artificial neural network with binary neurons. It is characterized by:

- Binary neurons with states +1 or -1.
- Symmetric weight matrix with zero diagonal (no self-connections).
- Energy function that is minimized at stable states.
- Asynchronous update of neuron states.

**Dynamics**

The state of each neuron is updated according to:

$$s_i(t + 1) = \text{sign} \left( \sum_j w_{ij} s_j(t) \right) \tag{16}$$

Where $s_i(t+1)$ is the state of neuron $i$ at time $t+1$, $w_{ij}$ is the weight between neurons $i$ and $j$, and $s_j(t)$ is the state of neuron $j$ at time $t$.

The energy of the network is defined as:

$$E = -\frac{1}{2} \sum_{i,j} w_{ij} s_i s_j \tag{17}$$

**Memory Storage and Retrieval**

Memories are stored in the network by adjusting the weights to minimize the network energy, often using the Hebbian learning rule. The network can retrieve memory from a noisy or incomplete version by converging to a stored state. The memory capacity of a Hopfield network depends on several factors, with the most significant one being the number of neurons in the network. John Hopfield proposed a rule in his original paper to estimate the memory capacity, stating that the network can effectively store approximately 0.15N independent memories, where N represents the number of neurons in the network. This means that for a network containing 100 neurons, it can store approximately 15 patterns.

### A.2.2 Modern Hopfield Network

In order to integrate Hopfield networks into deep learning architectures, The Modern Hopfield Network allows for continuous state updating. It proposes a new energy function based on the associative memory model [7] and proposes a new update rule that can be proven to converge to stationary points of the energy (local minima or saddle points). Specifically, the new Energy function is:

$$\text{E} = -\operatorname{lse}\left(\beta, \boldsymbol{X}^T \boldsymbol{\xi}\right) + \frac{1}{2} \boldsymbol{\xi}^T \boldsymbol{\xi} + \beta^{-1} \log N + \frac{1}{2} M^2, \tag{18}$$

with $lse(log-sum-exp)$ interaction function:

$$\operatorname{lse}(\beta, \boldsymbol{x}) = \beta^{-1} \log \left( \sum_{i=1}^{N} \exp\left(\beta x_i\right) \right) \tag{19}$$

Sering $0\text{E}2M^2$. Using $\boldsymbol{p} = \operatorname{softmax}\left(\beta \boldsymbol{X}^T \boldsymbol{\xi}\right)$, The novel update rule is:

$$\boldsymbol{\xi}^{\text{new}} = f(\boldsymbol{\xi}) = \boldsymbol{X} \boldsymbol{p} = \boldsymbol{X} \operatorname{softmax}\left(\beta \boldsymbol{X}^T \boldsymbol{\xi}\right). \tag{20}$$

The new update rule can be viewed as the attention in Transformers. Firstly, $N$ stored (key) patterns $\boldsymbol{y}_i$ and $S$ state (query) patterns $\boldsymbol{r}_i$ that are mapped to the Hopfield space of dimension $d_k$. Then set $\boldsymbol{x}_i = \boldsymbol{W}_K^T \boldsymbol{y}_i, \boldsymbol{\xi}_i = \boldsymbol{W}_Q^T \boldsymbol{r}_i$, and multiply the result of the update rule with $\boldsymbol{W}_V$. The matrices $\boldsymbol{Y} = (\boldsymbol{y}_1, \ldots, \boldsymbol{y}_N)^T$ and $\boldsymbol{R} = (\boldsymbol{r}_1, \ldots, \boldsymbol{r}_S)^T$ combine the $\boldsymbol{y}_i$ and $\boldsymbol{r}_i$ as row vectors. By defining the matrices $\boldsymbol{X}^T = \boldsymbol{K} = \boldsymbol{Y} \boldsymbol{W}_K, \boldsymbol{\Xi}^T = \boldsymbol{Q} = \boldsymbol{R} \boldsymbol{W}_Q$, and $\boldsymbol{V} = \boldsymbol{Y} \boldsymbol{W}_K \boldsymbol{W}_V = \boldsymbol{X}^T \boldsymbol{W}_V$, where $\boldsymbol{W}_K \in \mathbb{R}^{d_y \times d_k}, \boldsymbol{W}_Q \in \mathbb{R}^{d_r \times d_k}, \boldsymbol{W}_V \in \mathbb{R}^{d_k \times d_v}$, $\beta = 1/\sqrt{d_k}$ and softmax $\in \mathbb{R}^N$ is changed to a row vector, the update rule multiplied by $\boldsymbol{W}_V$ is:

$$\boldsymbol{Z} = \operatorname{softmax}\left(1/\sqrt{d_k} \boldsymbol{Q} \boldsymbol{K}^T\right) \boldsymbol{V} = \operatorname{softmax}\left(\beta \boldsymbol{R} \boldsymbol{W}_Q \boldsymbol{W}_K^T \boldsymbol{Y}^T\right) \boldsymbol{Y} \boldsymbol{W}_K \boldsymbol{W}_V.$$

In the Hopfield Evolution strategy of Attraos, The Query is settled as the dynamic structures and the Key/Value is settled as trainable vectors.

### A.3 Orthogonal Polynomials

**Note**: In this section, we have selectively extracted key content from the appendix of Hippo [11] that is pertinent to our work, for the convenience of the reader. We take Legendre polynomials as an example.

Under the usual definition of the canonical Legendre polynomial $P_n$, they are orthogonal with respect to the measure $\omega^{\text{leg}} = \mathbf{1}_{[-1,1]}$ :

$$\frac{2n+1}{2} \int_{-1}^{1} P_n(x) P_m(x) \mathrm{d}x = \delta_{nm}$$

Also, they satisfy

$$P_n(1) = 1$$
$$P_n(-1) = (-1)^n.$$

With respect to the measure $\frac{1}{\theta}\mathbb{I}[t - \theta, t]$, the normalized orthogonal polynomials are

$$(2n + 1)^{1/2} P_n\left(2\frac{x - t}{\theta} + 1\right)$$

In general, the orthonormal basis for any uniform measure consists of $(2n + 1)^{\frac{1}{2}}$ times the corresponding linearly shifted version of $P_n$.

**Derivatives of Legendre polynomials** We note the following recurrence relations on Legendre polynomials:

$$(2n + 1)P_n = P'_{n+1} - P'_{n-1}$$
$$P'_{n+1} = (n + 1)P_n + xP'_n$$

The first equation yields

$$P'_{n+1} = (2n + 1)P_n + (2n - 3)P_{n-2} + \dots,$$

where the sum stops at $P_0$ or $P_1$. These equations directly imply

$$P'_n = (2n - 1)P_{n-1} + (2n - 5)P_{n-3} + \dots$$

and

$$(x + 1)P'_n(x) = P'_{n+1} + P'_n - (n + 1)P_n$$
$$= nP_n + (2n - 1)P_{n-1} + (2n - 3)P_{n-2} + \dots$$

To sum up, The Legendre polynomials are in closed-recursive form.

# B   Proof

**Proposition 5.** *[18] $A = \mathrm{diag}\{-1, -1, \dots\}$ is a rough approximation of shifted Hippo-LegT [11] matrix.*

*Proof.* Hippo 3 provides a mathematical framework for deriving the $\boldsymbol{AB}$ matrix for polynomial projection. The mainstream SSMs [10] typically initialize matrix $\boldsymbol{A} = \mathrm{diag}\{-1, -2, -3, -4, \dots\}$, representing the negative real diagonal elements of the normalized Hippo-LegS matrix (21). Since the LegS matrix is a mathematical approximation of exponentially decaying Legendre polynomials, initializing $\boldsymbol{A} = \mathrm{diag}\{-1, -2, -3, -4, \dots\}$ can be seen as a rough approximation of exponential decay. Similarly, for the normalized Hippo-LegT matrix (22), which approximates a uniform measure, we can consider $\boldsymbol{A} = \mathrm{diag}\{-1, -1, \dots\}$ as a rough approximation of a finite window of Legendre polynomials, which better for non-stationary time series as well [19]. The use of negative values for the elements is to ensure gradient stability during training.

$$\boldsymbol{A}_{nk}^{(N)} = -\begin{cases} (n + \frac{1}{2})^{1/2}(k + \frac{1}{2})^{1/2} & n > k \\ \frac{1}{2} & n = k \\ (n + \frac{1}{2})^{1/2}(k + \frac{1}{2})^{1/2} & n < k \end{cases} \qquad \boldsymbol{A}_{nk}^{(N)} = -\begin{cases} (2n + 1)^{\frac{1}{2}}(2k + 1)^{\frac{1}{2}} & n < k, k \text{ odd} \\ 0 & else \\ (2n + 1)^{\frac{1}{2}}(2k + 1)^{\frac{1}{2}} & n > k, n \text{ odd} \end{cases}$$

$$\boldsymbol{A} = \boldsymbol{A}^{(N)} - \mathrm{rank}(1), \quad \boldsymbol{A}^{(D)} := \mathrm{eig}(\boldsymbol{A}^{(N)}) \qquad \boldsymbol{A} = \boldsymbol{A}^{(N)} - \mathrm{rank}(2), \quad \boldsymbol{A}^{(D)} := \mathrm{eig}(\boldsymbol{A}^{(N)})$$

$$\text{(Normal / DPLR form of HiPPO-LegS)} \qquad\qquad \text{(Normal / DPLR form of HiPPO-LegT)}$$

$$(21) \qquad\qquad\qquad\qquad\qquad\qquad\qquad\qquad (22)$$

$\square$

**Theorem 6.** *(Approximation Error Bound) Suppose that the function $f : [0, 1] \in \mathbb{R}$ is $k$ times continuously differentiable, the piecewise polynomial $g$ approximates $f$ with mean error bounded as follows:*

$$\|f - g\| \le 2^{-rk}\frac{2}{4^N k!} \sup_{x \in [0,1]} \left|f^{(k)}(x)\right|.$$

*Proof.* Similar to [1], we divide the interval $[0, 1]$ into subintervals on which $g$ is a polynomial; the restriction of $g$ to one such subinterval $I_{r,l}$ is the polynomial of degree less than $k$ that approximates $f$ with minimum mean error. Also, the optimal $g$ can be regarded as the orthonormal projection $Q_r^N f$ onto $\mathcal{G}_{(r)}^N$. We then use the maximum error estimate for the polynomial, which interpolates $f$ at Chebyshev nodes of order $k$ on $I_{r,l}$. We define $I_{r,l} = [2^{-r}l, 2^{-r}(l+1)]$ for $l = 0, 1, \ldots, 2^r - 1$, and obtain

$$
\begin{aligned}
\left\| Q_r^N f - f \right\|^2 &= \int_0^1 \left[ \left( Q_r^N f \right)(x) - f(x) \right]^2 dx \\
&= \sum_l \int_{I_{r,l}} \left[ \left( Q_r^N f \right)(x) - f(x) \right]^2 dx \\
&\leq \sum_l \int_{I_{r,l}} \left[ \left( C_{r,l}^N f \right)(x) - f(x) \right]^2 dx \\
&\leq \sum_l \int_{I_{r,l}} \left( \frac{2^{1-rk}}{4^N k!} \sup_{x \in I_{r,l}} \left| f^{(k)}(x) \right| \right)^2 dx \\
&\leq \left( \frac{2^{1-rk}}{4^N k!} \sup_{x \in [0,1]} \left| f^{(k)}(x) \right| \right)^2
\end{aligned}
$$

and by taking square roots we have bound (7). Here $C_{r,l}^N f$ denotes the polynomial of degree $k$ which agrees with $f$ at the Chebyshev nodes of order $k$ on $I_{r,l}$, and we have used the well-known maximum error bound for Chebyshev interpolation.

The error of the approximation $Q_r^N f$ of $f$ therefore decays like $2^{-rk}$ and, since $S_r^N$ has a basis of $2^r k$ elements, we have convergence of order $k$. For the generalization to $m$ dimensions in the dynamic structure modeling, a similar argument shows that the rate of convergence is of order $k/m$. $\qquad\square$

**Theorem 7.** *(**Evolution Error Bound**) By Jacobian value, the mean attractor evolution error $\left\| \mathcal{K} \circ \tilde{\mathcal{A}} - \tilde{\mathcal{A}} \right\|$ of evolution operator $\mathcal{K} = \{\mathcal{K}^{(i)}\}$ is bounded by*

$$
\left\| \mathcal{K} \circ \tilde{\mathcal{A}} - \tilde{\mathcal{A}} \right\| (N-1) \mathcal{E} \left( -\beta \nabla_i + 1 \right)
$$

*with the number of random patterns $N \geq \sqrt{p} c \frac{d-1}{4}$ stored in the system by interaction paradigm $\mathcal{E}$ in an ideal spherical retrieval paradigm.*

*Proof.* Due to the numerous assumptions and extensive lemmas involved in the proof of this theorem, we will provide a brief exposition of its main ideas. For a detailed proof, please refer to [38].

Firstly, the theorem defines the matching between patterns as:

**Definition 1.** *(Pattern match). Assuming that around every pattern $\boldsymbol{x}_i$ a sphere $S_i$ is given. We say $\boldsymbol{x}_i$ is matched with $\boldsymbol{\xi}$ if there is a single fixed point $\boldsymbol{x}_i^* \in S_i$ to which all points $\boldsymbol{\xi} \in S_i$ converge.*

As shown in **Theorem 3** in [38], according to the upper branch of the Lambert $W$ function [35], we can obtain the number of random patterns stored in a system is $N \geq \sqrt{p} c \frac{d-1}{4}$.

In our paper, we define the pattern1 as the Attractor $\tilde{\mathcal{A}}$ in phase space, pattern2 as the future state evaluated by operator $\mathcal{K}$, noted as $\mathcal{K} \circ \tilde{\mathcal{A}}$. From **Lemma A4** in [38], when the radius of the pattern matching sphere is $M$, we can describe the evolution process by jacobian value and get the matching error as:

$$
\left\| \mathcal{K} \circ \tilde{\mathcal{A}} - \tilde{\mathcal{A}} \right\| 2\epsilon M
$$

where in the **Equation (179)** of [38]:

$$
\epsilon = (N-1) \exp \left( -\beta \left( \nabla_i - 2 \max \left\{ \left\| \mathcal{K} \circ \tilde{\mathcal{A}} - \tilde{\mathcal{A}} \right\|, \left\| \tilde{\mathcal{A}}_i^* - \tilde{\mathcal{A}}_i \right\| \right\} M \right) \right).
$$

The **Equation (404)** of [38] says $\left\| \tilde{\mathcal{A}}_i^* - \tilde{\mathcal{A}}_i \right\| \frac{1}{2\beta M}$ and $\left\| \mathcal{K} \circ \tilde{\mathcal{A}} - \tilde{\mathcal{A}} \right\| \frac{1}{2\beta M}$, so we can get:

$$
\epsilon e (N-1) M \exp \left( -\beta \nabla_i \right).
$$

In our paper, we replace the exponential interaction function with an unknown function $\mathcal{E}(\cdot)$ to finally get the version in Theorem 3.

$\square$

**Theorem 8.** *(Completeness in $L^2$ space)* *The orthonormal system* $B_k = \{\phi_j : j = 1, \ldots, k\} \cup \{\psi_j^{rl} : j = 1, \ldots, k; r = 0, 1, 2, \ldots; l = 0, \ldots, 2^r - 1\}$ *spans* $L^2[0, 1]$.

*Proof.* We define the space $\mathcal{G}^N$ to be the union of the $\mathcal{G}_{(r)}^N$, given by the formula:

$$\mathcal{G}^N = \bigcup_{r=0}^{\infty} \mathcal{G}_r^N \tag{23}$$

and observe that $\overline{\mathcal{G}^N} = L^2[0, 1]$. In particular, $\mathcal{G}^N$ contains the Haar basis for $L^2[0, 1]$, consisting of functions piecewise constant on each of the subintervals $(2^{-r}l, 2^{-r}(l+1))$. Here the closure $\overline{\mathcal{G}^N}$ is defined with respect to the $L^2$-norm,

$$\|f\| = \langle f, f \rangle^{1/2}$$

where the inner product $\langle f, g \rangle$ is defined by the formula

$$\langle f, g \rangle = \int_0^1 f(x)g(x)dx.$$

Also, we have:

$$\mathcal{G}_r^N = \mathcal{G}_0^N \oplus \mathcal{S}_0^N \oplus \mathcal{S}_1^N \oplus \cdots \oplus \mathcal{S}_{r-1}^N \tag{24}$$

and

$$\mathcal{S}_r^N = \text{ linear span } \left\{ \psi_{j,r}^l : \psi_{j,r}^l(x) = 2^{r/2}\psi_j\left(2^r x - l\right), j = 1, \ldots, k; n = 0, \ldots, 2^r - 1 \right\}. \tag{25}$$

We let $\{\phi_1, \ldots, \phi_k\}$ denote an orthonormal basis for $\mathcal{G}_0^N$; in view of Equation 23, 24, and 25, the orthonormal system

$$B_k = \begin{array}{ll} \{\phi_j : & j = 1, \ldots, k\} \\ \cup \{h_{j,r}^l : & j = 1, \ldots, k; r = 0, 1, 2, \ldots; l = 0, \ldots, 2^r - 1\} \end{array}$$

spans $L^2[0, 1]$.

Now we construct a basis for $L^2(\mathbf{R})$ by defining, for $r \in \mathbf{Z}$, the space $\tilde{\mathcal{G}}_r^N$ by the formula $\tilde{\mathcal{G}}_r^N = \{f :$ the restriction of $f$ to the interval $(2^{-r}n, 2^{-r}(n+1))$ is a polynomial of degree less than $k$, for $n \in \mathbf{Z}\}$ and observing that the space $\tilde{\mathcal{G}}_{r+1}^N \backslash \tilde{\mathcal{G}}_r^N$ is spanned by the orthonormal set

$$\left\{ \psi_{j,r}^l : \quad \psi_{j,r}^l(x) = 2^{r/2}\psi_j\left(2^r x - l\right), j = 1, \ldots, k; l \in \mathbf{Z} \right\}.$$

Thus $L^2(\mathbf{R})$, which is contained in $\overline{\bigcup_r \tilde{\mathcal{G}}_r^N}$, has an orthonormal basis

$$\left\{ \psi_{j,m}^n : j = 1, \ldots, k; r, l \in \mathbf{Z} \right\}$$

$\square$

## C   Model Details

### C.1   Phase Space Reconstruction

Phase space reconstruction is a crucial technique in the analysis of dynamical systems, particularly in the study of time series data. This method transforms a one-dimensional time series into a multidimensional phase space, revealing the underlying dynamics of the system. The cross-correlation mutual

information (CC mutual information) method is a statistical tool used to analyze the dependencies between two different time series. We provide a detailed mathematical description of these methods.

Phase space reconstruction involves transforming a single-variable time series into a multidimensional space to unveil the dynamics of the system that generated the data. The technique is based on Takens' Embedding Theorem.

**Takens' Embedding Theorem**

Takens' Embedding Theorem allows the reconstruction of a dynamical system's phase space from a sequence of observations. The theorem states that under generic conditions, a map:

$$F : \mathbb{R}^n \to \mathbb{R}^m \tag{26}$$

can be constructed, where $n$ is the dimension of the original phase space, and $m$ is the embedding dimension, typically $m <= 2n + 1$ is sufficient to recover dynamics.

**Time Delay Embedding**

The most common approach for phase space reconstruction is the time delay embedding method. Given a time series $\{x(t)\}$, the reconstructed phase space is:

$$X(t) = [x(t), x(t + \tau), x(t + 2\tau), ..., x(t + (m - 1)\tau)] \tag{27}$$

where $\tau$ is the time delay, and $m$ is the embedding dimension. In the context of coordinate delay reconstruction in a complete scenario, it involves a parameter called time window, which selectively uses discrete one-dimensional data for phase space reconstruction. In order to maximize the preservation of time series information, the time window parameter is typically set to 1. As a result, this process is reversible, meaning that the original data can be reconstructed without loss of information.

The CC mutual information method is used to analyze the dependencies between two time series. It is a measure of the amount of information obtained about one time series through the other.

**Mutual Information**

Mutual information $I(X;Y)$ between two random variables $X$ and $Y$ is defined as:

$$I(X;Y) = \sum_{x \in X, y \in Y} p(x, y) \log \frac{p(x, y)}{p(x)p(y)} \tag{28}$$

where $p(x, y)$ is the joint probability distribution function of $X$ and $Y$, and $p(x)$ and $p(y)$ are the marginal probability distribution functions.

**Cross-Correlation Mutual Information**

For time series analysis, the mutual information is extended to account for the time-lagged relationships:

$$I(X;Y,\tau) = \sum_{x \in X, y \in Y} p(x, y(\tau)) \log \frac{p(x, y(\tau))}{p(x)p(y(\tau))} \tag{29}$$

where $\tau$ is the time lag, and $y(\tau)$ represents the time series $Y$ shifted by $\tau$.

**Determining Time Delay ($\tau$)**

Time delay ($\tau$) is an interval used in reconstructing the phase space of a time series. The right choice of $\tau$ is essential for revealing the dynamic properties of the system.

**Calculating Mutual Information**

For a given time series $\{x_t\}$, calculate the mutual information between the time series and its time-shifted version for different delays $\tau$:

$$I(\tau) = \sum p(x_t, x_{t+\tau}) \log \left( \frac{p(x_t, x_{t+\tau})}{p(x_t)p(x_{t+\tau})} \right) \tag{30}$$

where $p(x_t, x_{t+\tau})$ is the joint probability distribution, and $p(x_t)$ and $p(x_{t+\tau})$ are the marginal probability distributions.

**Selecting Time Delay**

Plot the mutual information $I(\tau)$ against $\tau$. Choose the $\tau$ at the first local minimum of this plot. This represents the delay where the series' points provide maximal mutual information.

**Determining Embedding Dimension ($m$)**

Embedding dimension ($m$) is the dimension of the reconstructed phase space. The correct $m$ ensures that trajectories in the phase space do not intersect each other.

**Calculating False Nearest Neighbors**

For each dimension $m$, calculate the false nearest neighbors (FNN) $F(m)$, which reflects the complexity of the trajectories reconstructed in $m$-dimensional space:

$$F(m) = \left( \frac{1}{N-m+1} \sum_{i=1}^{N-m+1} \log C_i(m, r) \right) \tag{31}$$

where $C_i(m, r)$ is the count of points within a distance $r$ from point $i$ in $m$-dimensional space, and $N$ is the length of the time series.

**Identifying the Saturation Point**

As $m$ increases, $F(m)$ typically increases and reaches a saturation point. Choose the smallest $m$ for which $F(m)$ does not significantly increase, indicating that increasing the dimension does not reveal more information about the dynamics.

# D    Experiments Details

## D.1    Datasets

Our experiments are carried out on Five real-world datasets and two chaos datasets as described below:

- **ETT**[2] dataset are procured from two electricity substations over two years. They provide a summary of load and oil temperature data across seven variables. For ETTm1 and ETTm2, the "m" signifies that data was recorded every 15 minutes, yielding 69,680 time steps. ETTh1 and ETTh2 represent the hourly equivalents of ETTm1 and ETTm2, each containing 17,420 time steps.

- **Exchange**[3] dataset details the daily foreign exchange rates of eight countries, including Australia, British, Canada, Switzerland, China, Japan, New Zealand, and Singapore from 1990 to 2016.

- **Weather**[4] dataset is a meteorological collection featuring 21 variates, gathered in the United States over a four-year span.

- **Lorenz96** dataset is an artificial dataset. We simulate the data of 30,000 time steps with an initial dimension of 40d according to the Lorenz96 equation and map to 3D by a randomly initialized Linear network to simulate the realistic chaotic time series generated from an unknown underlying chaotic system. More details are in Appendix E.3.

- **Crypots** comprises historical transaction data for various cryptocurrencies, including Bitcoin and Ethereum. We select samples with $Asset\_ID$ set to 0, and remove the column $Count$. We download the data from `https://www.kaggle.com/competitions/g-research-crypto-forecasting/data?select=supplemental_train.csv`.

- **Air Convection**[5] dataset is obtained by scraping the data from NOAA, and it includes the data for the entire year of 2023. The dataset consists of 20 variables, including air humidity, pressure, convection characteristics, and others. The data was sampled at intervals of 15 minutes and averaged over the course of the entire year.

---

[2]https://github.com/zhouhaoyi/ETDataset

[3]https://github.com/laiguokun/multivariate-time-series-data

[4]https://www.bgc-jena.mpg.de/wetter

[5]https://www.psl.noaa.gov/

- **Lorenz96-3d**: See Appendix E.3.

## D.2 Baselines

Our baseline models include: **Mamba4TS:** Mamba4TS is a novel SSM architecture tailored for TSF tasks, featuring a parallel scan (https://github.com/alxndrTL/mamba.py/tree/main). Additionally, this model adopts a patching operation with both patch length and stride set to 16. We use the recommended configuration as our experimental settings with a batch size of 32, and the learning rate is 0.0001.

**S-Mamba [47]:** S-Mamba utilizes a linear tokenization of variates and a bidirectional Mamba layer to efficiently capture inter-variate correlations and temporal dependencies. This approach underscores its potential as a scalable alternative to Transformer technologies in TSF. We download the source code from: https://github.com/wzhwzhwzh0921/S-D-Mamba and adopt the recommended setting as its experimental configuration.

**RWKV-TS [17]:** RWKV-TS is an innovative RNN-based architecture for TSF that offers linear time and memory efficiency. We download the source code from: https://github.com/howard-hou/RWKV-TS. We follow the recommended settings as experimental configuration.

**Koopa [32]:** Koopa is a novel forecasting model that tackles non-stationary time series using Koopman theory to differentiate time-variant dynamics. It features a Fourier Filter and Koopman Predictors within a stackable block architecture, optimizing hierarchical dynamics learning. We download the source code from: https://github.com/thuml/Koopa. We set the lookback window to fixed values of {96, 192, 336, 720} instead of twice the output length as in the original experimental settings.

**iTransformer [31]:** iTransformer modifies traditional Transformer models for time series forecasting by inverting dimensions and applying attention and feed-forward networks across variate tokens. We download the source code from: https://github.com/thuml/iTransformer. We follow the recommended settings as experimental configuration.

**PatchTST [33]:** PatchTST introduces a novel design for Transformer-based models tailored to time series forecasting. It incorporates two essential components: patching and channel-independent structure. We obtain the source code from: https://github.com/PatchTST. This code serves as our baseline for long-term forecasting, and we follow the recommended settings for our experiments.

**LTSF-Linear [52]:** In LTSF-Linear family, DLinear decomposes raw data into trend and seasonal components and NLinear is just a single linear models to capture temporal relationships between input and output sequences. We obtain the source code from: https://github.com/cure-lab/LTSF-Linear, using it as our long-term forecasting baseline and adhering to recommended settings for experimental configuration.

**GPT4TS [55]:** This study explores the application of pre-trained language models to time series analysis tasks, demonstrating that the Frozen Pretrained Transformer (FPT), without modifications to its core architecture, achieves state-of-the-art results across various tasks. We download the source code from: https://github.com/DAMO-DI-ML/NeurIPS2023-One-Fits-All. We follow the recommended settings as experimental configuration.

## D.3 Experiments Setting

All experiments are conducted on the NVIDIA RTX3090-24G and A6000-48G GPUs. The Adam optimizer is chosen. A grid search is performed to determine the optimal hyperparameters, including the learning rate from {0.0001, 0.0005, 0.001}, model layer from {1, 2} (typically 1), polynomial order from {32, 64, 256}, patch length from {8, 16}, projection level based on $logL$ and no more than 3, primary modes is 16, and the input length is 96. Due to the sensitivity of the CC method to numerical calculations, we take the average result from three iterations of the calculation.

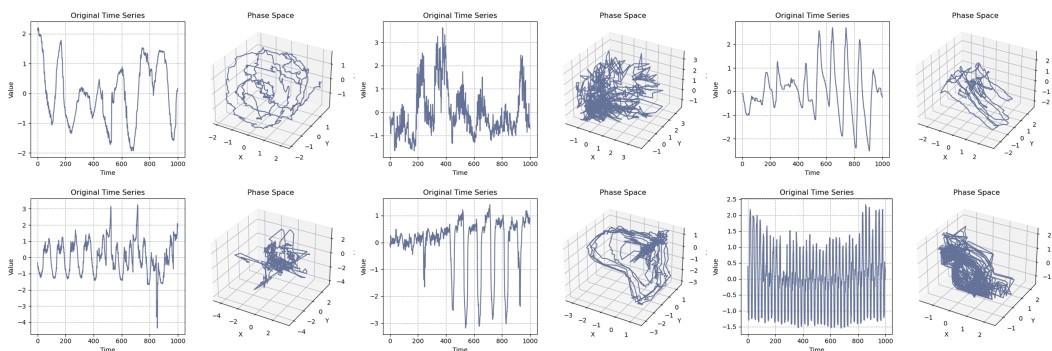

Figure 6: Dynamic structures of real-world data

# E  Supplemental Experiments

## E.1  Dynamic Structures of Real-world Data

As shown in Figure 6, we present the phase space structures of various real-time series. Please note that due to our visualization limitations in three dimensions, the shapes of many attractors for these time series are manifested in higher dimensions. We can only display slices of the attractors in the first three dimensions.

## E.2  The Chaotic Nature of Datasets

### Lyaponuv Exponent

Lyapunov exponents are a set of quantities that characterize the rate of separation of infinitesimally close trajectories in a dynamical system. They play a crucial role in understanding the stability and chaotic behavior of the system [24]. The formal definition of the Lyapunov exponent for a trajectory of a dynamical system is given by:

$$\lambda = \lim_{t \to \infty} \frac{1}{t} \ln \left( \frac{\|\delta X_i(t)\|}{\|\delta X_i(0)\|} \right), \tag{32}$$

where $\delta \mathbf{X}(t)$ is the deviation vector at time $t$, and $\lambda$ is the Lyapunov exponent. The set of Lyapunov exponents for a system provides a spectrum, indicating the behavior of trajectories in each dimension of the system's phase space.

When the direction of the initial separation vector is different, the separation rate is also different. Thus, the spectrum of Lyapunov exponents exists, which have the same number of dimensions as the phase space. The largest of these is often referred to as the Maximal Lyapunov exponent (MLE). We reconstruct dimension $m$ according to different phase spaces to get the maximum Lyapunov index. A positive MLE is often considered an indicator of chaotic behavior in the system, while a negative value indicates convergence, implying stability in the system's behavior.

As shown in Table 6, we calculate the maximum Lyapunov exponent for all mainstream LTSF datasets. Surprisingly, we find that their MLEs are all positive, indicating the presence of chaos to varying degrees in these datasets. This directly supports the motivation proposed by Attraos. Among them, the weather dataset exhibits the strongest chaotic behavior. Note that the existence of at least one positive Lyapunov exponent is sufficient to determine the presence of chaos. For example, the classical Lorenz63 system has three Lyapunov exponents with values negative, zero, and positive respectively.

**Remark E.1.** *For multivariate time series, we take the average.*

Table 6: Maximal Lyapunov Exponents for Various Datasets in Multivariate Long-Term Forecasting

| Dataset | ETTh1 | ETTh2 | ETTm1 | ETTm2 | Exchange | Weather | Electricity | Traffic |
|---------|-------|-------|-------|-------|----------|---------|-------------|---------|
| MLE | 0.064437 | 0.059833 | 0.071673 | 0.082791 | 0.039670 | 0.242649 | 0.014613 | 0.189311 |

### E.3 Simulation for Lorenz96 (Case Study)

**Lorenz63 System**

The Lorenz63 system, introduced by Edward Lorenz in 1963, is a simplified mathematical model for atmospheric convection. The model is a system of three ordinary differential equations now known as the Lorenz equations:

$$\frac{dx}{dt} = \sigma(y - x), \tag{33}$$

$$\frac{dy}{dt} = x(\rho - z) - y, \tag{34}$$

$$\frac{dz}{dt} = xy - \beta z. \tag{35}$$

Here, $x$, $y$, and $z$ make up the state of the system. The parameters $\sigma$, $\rho$, and $\beta$ represent the Prandtl number, Rayleigh number, and certain physical dimensions of the layer, respectively. Lorenz derived these equations to model the way air moves around in the atmosphere. This model is famous for exhibiting chaotic behavior for certain parameter values and initial conditions.

**Lorenz96 System**

The Lorenz96 system is a mathematical model that was introduced by Edward N. Lorenz in 1996 as an extension of the original Lorenz model. It is commonly used to study chaotic dynamics in systems with multiple interacting variables.

The Lorenz96 system consists of a set of ordinary differential equations that describe the time evolution of a set of variables. In its simplest form, the system is defined as follows:

$$\frac{dx_i}{dt} = (x_{i+1} - x_{i-2})x_{i-1} - x_i + F$$

Here, $x_i$ represents the state variable at position $i$, and $F$ is a forcing term that controls the overall behavior of the system. The nonlinear term $(x_{i+1} - x_{i-2})x_{i-1}$ captures the interactions between neighboring variables, leading to the emergence of chaotic behavior. The Lorenz96 system exhibits a range of fascinating phenomena, including intermittent chaos, phase transitions, and the presence of multiple stable and unstable regimes. Its dynamics have been extensively studied to gain insights into the behavior of complex systems and to explore the limits of predictability.

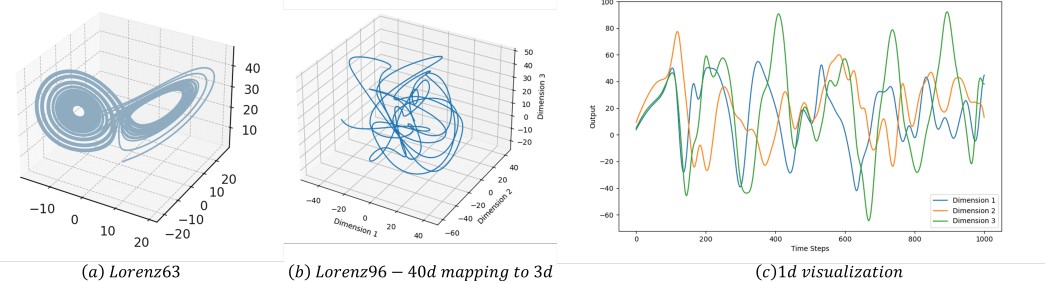

$(a)$ *Lorenz63*      $(b)$ *Lorenz96 − 40d mapping to 3d*      $(c)$ *1d visualization*

Figure 7: Simulation for Lorenz96

To simulate time series generated from an unknown chaotic system, we first construct a 40-dimensional Lorenz96 system to represent the underlying chaotic system. We then use a randomly initialized linear neural network to simulate the observation function $h$. Through $h$, we map the Lorenz96-40d system, which resides in the manifold space, to a 3-dimensional Euclidean space to obtain a multivariate time series dataset with three variables. As shown in the middle of Figure 7, in the observation domain, the dynamical system structure of Lorenz96-40d is difficult to discern specific shapes, so we need to reconstruct it to 40 dimensions using the Phase Space Reconstruction (PSR) technique to study its dynamic characteristics in a topologically equivalent structure. It is worth noting that on the right side of Figure 7, we visualize the last variable of the system and find striking similarities with real-world time series, even exhibiting some periodic behaviors. This further supports our hypothesis that real-world time series are generated by underlying chaotic systems.

## E.4 Chaotic Modulation

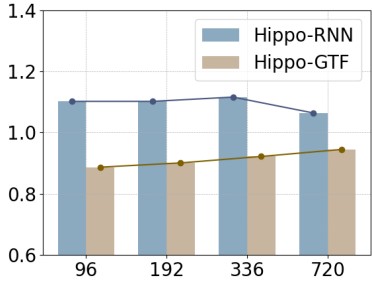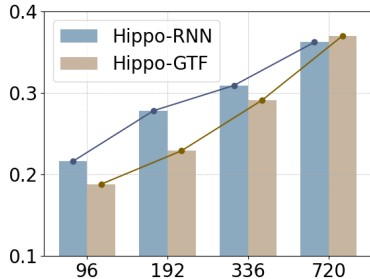

Figure 8: Performance comparison about teaching forcing, measured by MSE. Left: Lorenzo96 dataset. Right: Weather dataset.

In LTSF tasks, it is commonly observed that predictive performance deteriorates as the length of the forecasting window increases. This phenomenon bears a striking resemblance to the inherent challenge in long-term predictions of chaotic dynamical systems, which are highly sensitive to initial conditions. Recent studies [15] in the field of chaotic dynamical systems have highlighted that, to address the issue of gradient divergence caused by positive maximal Lyapunov exponents indicative of chaos, the implementation of teaching forcing as a method to incrementally constrain the trajectory of the dynamical system presents a straightforward yet effective framework. To enhance Hippo-RNN, we have implemented the following modifications: $\tilde{z}_t := (1 - \alpha)z_t + \alpha\overline{z}_t,\ z_t = F_{\theta}\left(\tilde{z}_{t-1}\right),$ with $0 \leq \alpha \leq 1$, where we intervene the evolution state $z$ by utilizing the ground truth hidden state $\overline{z}$. As shown in Figure 8, this modification leads to notable improvements in both mainstream LTSF datasets and real-world chaotic datasets. However, the autoregressive nature of the teaching forcing methods introduces additional computational overhead and may potentially reduce its generalization capabilities, which presents a challenge for integrating it into sequence-mapping models.

## E.5 Full Results

As shown in Table 7, we showcase the full experiment results for all mainstream LTSF datasets.

Table 7: Multivariate long-term series forecasting results in mainstream datasets with input length is 96 and prediction horizons are {96, 192, 336, 720}. The best model is in boldface, and the second best is underlined. The bottom part introduces the variable Kernel for multivariate variables.

| | Model | Attraos (Ours) | | Mamba4TS (Temporal Emb.) | | S-Mamba (Channel Emb.[47]) | | RWKV-TS [17] | | GPT4TS [55] | | Koopa [32] | | InvTrm [31] | | PatchTST [33] | | DLinear [52] | |
|---|---|---|---|---|---|---|---|---|---|---|---|---|---|---|---|---|---|---|---|
| | Metric | MSE | MAE | MSE | MAE | MSE | MAE | MSE | MAE | MSE | MAE | MSE | MAE | MSE | MAE | MSE | MAE | MSE | MAE |
| ETTh1 | 96 | **0.370** | **0.388** | 0.386 | 0.400 | 0.388 | 0.407 | 0.383 | 0.401 | 0.398 | 0.424 | 0.385 | 0.408 | 0.393 | 0.409 | 0.375 | 0.396 | 0.396 | 0.410 |
| | 192 | **0.416** | **0.418** | 0.426 | 0.430 | 0.443 | 0.439 | 0.441 | 0.431 | 0.441 | 0.436 | 0.441 | 0.431 | 0.448 | 0.442 | 0.429 | 0.426 | 0.449 | 0.444 |
| | 336 | **0.458** | **0.432** | 0.484 | 0.451 | 0.492 | 0.467 | 0.493 | 0.465 | 0.492 | 0.466 | 0.474 | 0.454 | 0.491 | 0.465 | 0.461 | 0.448 | 0.487 | 0.465 |
| | 720 | **0.447** | **0.442** | 0.481 | 0.472 | 0.511 | 0.499 | 0.501 | 0.487 | 0.487 | 0.483 | 0.501 | 0.480 | 0.518 | 0.501 | 0.469 | 0.471 | 0.515 | 0.512 |
| | AVG | **0.423** | **0.420** | 0.444 | 0.438 | 0.459 | 0.453 | 0.454 | 0.446 | 0.457 | 0.450 | 0.450 | 0.443 | 0.463 | 0.454 | 0.434 | 0.435 | 0.462 | 0.458 |
| ETTh2 | 96 | 0.292 | 0.348 | 0.297 | 0.347 | 0.296 | 0.347 | **0.290** | **0.342** | 0.312 | 0.360 | 0.317 | 0.359 | 0.302 | 0.351 | 0.295 | 0.344 | 0.353 | 0.405 |
| | 192 | 0.374 | **0.386** | 0.392 | 0.409 | 0.377 | 0.398 | **0.372** | 0.393 | 0.387 | 0.405 | 0.375 | 0.399 | 0.379 | 0.399 | 0.420 | 0.429 | 0.482 | 0.479 |
| | 336 | 0.420 | 0.432 | 0.424 | 0.436 | 0.425 | 0.435 | **0.417** | 0.431 | 0.424 | 0.437 | 0.436 | 0.446 | 0.423 | 0.432 | 0.420 | **0.429** | 0.588 | 0.539 |
| | 720 | **0.418** | **0.431** | 0.431 | 0.448 | 0.427 | 0.446 | 0.421 | 0.442 | 0.433 | 0.453 | 0.460 | 0.463 | 0.429 | 0.447 | 0.431 | 0.451 | 0.833 | 0.658 |
| | AVG | 0.376 | **0.399** | 0.386 | 0.410 | 0.381 | 0.407 | **0.375** | 0.402 | 0.389 | 0.414 | 0.397 | 0.417 | 0.383 | 0.407 | 0.380 | 0.406 | 0.564 | 0.520 |
| ETTm1 | 96 | **0.321** | 0.362 | 0.331 | 0.368 | 0.332 | 0.368 | 0.328 | 0.366 | 0.335 | 0.369 | 0.322 | **0.360** | 0.343 | 0.377 | 0.326 | 0.365 | 0.345 | 0.372 |
| | 192 | **0.365** | **0.373** | 0.376 | 0.391 | 0.378 | 0.393 | 0.372 | 0.389 | 0.374 | 0.385 | 0.378 | 0.393 | 0.379 | 0.394 | 0.361 | 0.383 | 0.382 | 0.391 |
| | 336 | **0.390** | **0.395** | 0.406 | 0.413 | 0.409 | 0.414 | 0.401 | 0.409 | 0.407 | 0.406 | 0.405 | 0.413 | 0.418 | 0.418 | 0.396 | 0.405 | 0.413 | 0.413 |
| | 720 | **0.451** | **0.432** | 0.469 | 0.452 | 0.476 | 0.453 | 0.462 | 0.446 | 0.469 | 0.442 | 0.473 | 0.447 | 0.488 | 0.458 | 0.458 | 0.439 | 0.472 | 0.450 |
| | AVG | **0.382** | **0.391** | 0.396 | 0.406 | 0.399 | 0.407 | 0.391 | 0.403 | 0.396 | 0.401 | 0.395 | 0.403 | 0.407 | 0.412 | 0.403 | 0.398 | 0.403 | 0.406 |
| ETTm2 | 96 | **0.172** | **0.254** | 0.186 | 0.268 | 0.182 | 0.267 | 0.181 | 0.264 | 0.190 | 0.275 | 0.180 | 0.261 | 0.184 | 0.269 | 0.177 | 0.260 | 0.192 | 0.291 |
| | 192 | **0.242** | **0.301** | 0.261 | 0.320 | 0.248 | 0.309 | 0.245 | 0.307 | 0.253 | 0.313 | 0.244 | 0.304 | 0.253 | 0.313 | 0.246 | 0.308 | 0.284 | 0.360 |
| | 336 | 0.303 | **0.340** | 0.331 | 0.366 | 0.312 | 0.350 | 0.306 | 0.344 | 0.321 | 0.360 | 0.313 | 0.340 | 0.313 | 0.351 | **0.302** | 0.343 | 0.371 | 0.420 |
| | 720 | 0.401 | **0.399** | 0.418 | 0.416 | 0.412 | 0.406 | 0.406 | 0.406 | 0.411 | 0.406 | **0.398** | 0.400 | 0.413 | 0.406 | 0.407 | 0.405 | 0.532 | 0.511 |
| | AVG | **0.280** | **0.324** | 0.299 | 0.343 | 0.289 | 0.333 | 0.285 | 0.330 | 0.294 | 0.339 | 0.281 | 0.326 | 0.291 | 0.335 | 0.283 | 0.329 | 0.345 | 0.396 |
| Exchange | 96 | **0.082** | **0.200** | 0.086 | 0.205 | 0.087 | 0.209 | 0.129 | 0.256 | 0.091 | 0.212 | 0.093 | 0.215 | 0.097 | 0.222 | 0.093 | 0.212 | 0.094 | 0.227 |
| | 192 | **0.171** | **0.294** | 0.173 | 0.297 | 0.180 | 0.303 | 0.231 | 0.346 | 0.183 | 0.304 | 0.189 | 0.313 | 0.184 | 0.309 | 0.201 | 0.319 | 0.185 | 0.325 |
| | 336 | 0.331 | 0.415 | 0.340 | 0.423 | 0.330 | 0.417 | 0.380 | 0.448 | 0.328 | 0.417 | 0.371 | 0.443 | **0.327** | **0.416** | 0.338 | 0.422 | 0.330 | 0.437 |
| | 720 | 0.810 | **0.669** | 0.855 | 0.696 | 0.860 | 0.700 | 0.883 | 0.704 | 0.880 | 0.704 | 0.908 | 0.726 | 0.885 | 0.715 | 0.900 | 0.711 | **0.774** | 0.673 |
| | AVG | 0.349 | **0.395** | 0.364 | 0.405 | 0.364 | 0.407 | 0.406 | 0.439 | 0.371 | 0.409 | 0.390 | 0.424 | 0.366 | 0.416 | 0.383 | 0.416 | **0.346** | 0.416 |
| Crypto | 96 | **0.174** | 0.143 | 0.179 | 0.143 | 0.187 | 0.147 | 0.176 | **0.139** | 0.183 | 0.144 | 0.181 | 0.143 | 0.183 | 0.144 | 0.177 | 0.141 | 0.183 | 0.155 |
| | 192 | **0.182** | **0.149** | 0.188 | 0.154 | 0.191 | 0.153 | 0.186 | 0.151 | 0.189 | 0.155 | 0.191 | 0.153 | 0.190 | 0.155 | 0.188 | 0.152 | 0.195 | 0.169 |
| | 336 | **0.191** | **0.158** | 0.197 | 0.166 | 0.197 | 0.163 | 0.192 | 0.162 | 0.201 | 0.168 | 0.208 | 0.173 | 0.199 | 0.167 | 0.195 | 0.164 | 0.206 | 0.180 |
| | 720 | **0.201** | **0.179** | 0.207 | 0.186 | 0.216 | 0.190 | 0.205 | 0.184 | 0.210 | 0.187 | 0.215 | 0.189 | 0.212 | 0.189 | 0.208 | 0.185 | 0.219 | 0.201 |
| | AVG | **0.187** | **0.157** | 0.193 | 0.162 | 0.198 | 0.163 | 0.190 | 0.159 | 0.196 | 0.164 | 0.199 | 0.165 | 0.196 | 0.164 | 0.192 | 0.161 | 0.201 | 0.176 |
| Weather | 96 | 0.159 | 0.206 | 0.175 | 0.215 | 0.165 | 0.208 | 0.175 | 0.217 | 0.203 | 0.244 | **0.158** | **0.203** | 0.175 | 0.216 | 0.176 | 0.217 | 0.197 | 0.259 |
| | 192 | 0.212 | 0.249 | 0.223 | 0.257 | 0.215 | 0.254 | 0.219 | 0.256 | 0.247 | 0.277 | **0.211** | **0.252** | 0.225 | 0.257 | 0.223 | 0.257 | 0.238 | 0.299 |
| | 336 | **0.265** | **0.288** | 0.278 | 0.297 | 0.273 | 0.297 | 0.275 | 0.298 | 0.297 | 0.311 | 0.267 | 0.292 | 0.280 | 0.298 | 0.277 | 0.296 | 0.282 | 0.331 |
| | 720 | **0.347** | **0.340** | 0.355 | 0.349 | 0.354 | 0.349 | 0.353 | 0.349 | 0.368 | 0.356 | 0.351 | 0.346 | 0.358 | 0.350 | 0.354 | 0.348 | 0.350 | 0.388 |
| | AVG | **0.246** | **0.271** | 0.258 | 0.280 | 0.252 | 0.277 | 0.256 | 0.280 | 0.279 | 0.279 | 0.247 | 0.273 | 0.260 | 0.280 | 0.258 | 0.280 | 0.267 | 0.319 |
| $1^{st}/2^{st}$Count | | **33** | **31** | 3 | 3 | 0 | 1 | **11** | 10 | 2 | 2 | 9 | 9 | 1 | 1 | 10 | 13 | 2 | 1 |

