# OpenReview forum: "Attractor Memory for Long-Term Time Series Forecasting: A Chaos Perspective"
_NeurIPS.cc/2024/Conference — NeurIPS 2024 poster_

### Official Review · Reviewer_qpQ4 · 2024-07-10

**Soundness:** 3
**Presentation:** 1
**Contribution:** 3
**Rating:** 6
**Confidence:** 2

**Summary:**

This paper introduces Attraos, a new model for long-term time series forecasting (LTSF) that incorporates chaos theory and views time series data as observations from high-dimensional chaotic dynamic systems. Attraos utilizes attractor invariance, non-parametric Phase Space Reconstruction, and a multi-scale dynamic memory unit to effectively capture historical dynamics and forecast future states with substantially fewer parameters than existing models such as PatchTST. Empirical evidence demonstrates Attraos' superior performance across various LTSF and chaotic datasets, providing a new perspective on the underlying dynamics of time series data.

**Strengths:**

-	The paper presents comprehensive experiment results and provide rigorous theoretical results to support their claims.
-	Leveraging results from chaotic theories for LTSF tasks is a novel and interesting idea.
-	The method proposed in the paper needs much fewer parameters than previous methods.

**Weaknesses:**

-	Most of the descriptions for problem setup (section 2), methods, and theoretical analysis (section 3) are very confusing and hard to follow. There are always notations suddenly coming out without much information. I suspect there are also several typos that cause difficulties in understanding and assessments. See **Questions** for details.
-	If I understood correctly, the title is a little misleading since seemingly the key insight/perspective comes from the embedding theorem by Takens, a counterpart of Whitney theorem for attractors. Are there any other insights related to chaotic systems? If not, I would suggest concretizing this point in the title, abstract, introduction, and conclusion.
-	Section A.1 in appendix is entirely copied from Wikipedia, https://en.wikipedia.org/wiki/Takens%27s_theorem , except for some changes on notations. However the notations in this section are not coherent, e.g. $k$ and $N$ in L414.
-	Similar to the first point, the paper does not provide enough background introduction.

If the authors worry containing all the necessary details will exceed the page limit, I would suggest presenting a detailed version of problem setup, backgrounds, analysis and methods in appendix.

**Questions:**

1.	L90 Is it a weighted integral $\mu(ds)$ instead of $ds$? For polynomials $\phi$, the integral with Lebesgue measure does not converge.
2.	Is $\mu$ same as $\omega$ in L94?
3.	Isn’t L94 giving the formula of projection to polynomial subspace and isn’t $K_n$ the kernel? How are they connected to $e^{tA}B$, where $A, B$ only appear in the induced dynamics of $x(t)$?
4.	Does $\mathcal{A}_i$ refer that there are multiple attractors in the dynamics? $\mathcal{A}$ are sets, then what does transpose in L158 stand for?
5.	L112, 120, if $A$ is a (D,N) tensor, $B$ is a (B,L,N) tensor and $u$ is a $(B,L,D)$ tensor, how is $Ax(t)+Bu(t)$ in (2a) defined?
6.	Could the authors give more interpretation on what they intend to do $\Delta$ and $\theta$? I have no idea what the authors mean by ‘$\Delta$ is similar to an attention mechanism’ and how this is related to the previous context.
7.	Again, given the shape of $B$ and $\Delta$, it is confusing what $\Delta B$ refers to.
8.	How is $H$ in L144 related to previous text?

**Limitations:**

Yes.

---

> ### Author Rebuttal · Authors · 2024-08-07
>
> Thank you very much for appreciating the technical novelty and efficiency achieved by our method. We are really sorry for missing several details, most of which delve into some details of the state-space model. Please allow us to provide you with a detailed response to questions 1-8.
> * **Q1: $\mu(ds)$ or $ds$?** Typically, describing two orthogonal bases involves the measure ($\mu(ds)$). However, we have opted for the simplified representation seen in prior state space model literature, omitting the measure when the integral context is evident. In the revised version of the paper, we will incorporate the complete $\mu(ds)$ and provide an explanation. Your observation is appreciated.
> * **Q2: The difference between $\mu$ and $\omega$:** In general, $\omega$ denotes the weight function, and $\mu(ds)= \omega(s) ds$. Expressing it in this manner within state space models enables the adaptable manipulation of basis and weight functions, facilitating the derivation of various state space model variants.
> * **Q3: The connection between $K(t,s)$ and $e^{tA}B$?** Equations 2a-2c (line 86) represent the standard equations for state space models [1-6]. These formulas utilize the kernel $K$ to signify the integral transformation. For a differential equation of $x^{\prime}(t)=Ax(t)+Bu(t)$, its general solution is:$$x(t)=e^{A (t-t_0)} x (t_0)+\int_{t_0}^t e^{A(t-s)}Bu(s) \mathrm{d}s.$$ When we assume the initial value is 0, i.e., $x(t_0)=0$, we can express $K(t,s)$ as $e^{tA}B$ based on the definition of matrix exponentiation.
> * **Q4: Details about $A$ and $\nabla$**: You are correct that an attractor for a dynamical system should be multiple, so in our paper, we use sets to represent this. The symbol $\nabla$ at line 158 signifies the distinctions between these attractor patterns. As per Theorem 3, it's crucial for $\nabla$ to be sufficiently large to maintain the stability of these attractor patterns during training (invariance). This necessity leads us to establish multiple orthogonal subspaces to store each $A_i$.
> * **Q5: The computation of $Ax+Bu$**: We have adhered to the tensor shape representation of the state space model (Mamba), and **in our open-source code, we have meticulously annotated the tensor shapes at each step of the model**. Specifically, Equation 2a represents the expression for continuous states. Given $A, B, \Delta$, we first obtain the discrete versions for actual computation using Equation 8 as $\overline{A}=e^{\Delta A}$ (B, L, D, N) and $\overline{B}=\Delta B$ (B, L, D, N). We then proceed with the calculations outlined in Formula 9 (also $x[k]=\overline{A}[k]* x[k-1]+ \overline{B}[k]*u[k], x[0]=0$ where the index is in L dimension).
> * **Q6: Details about $\Delta$ and $\theta$**:In the field of state space models, early studies such as LMU (NIPS 2019) and HIPPO (NIPS 2020) introduced the parameter $\theta$ to represent the size of the measurement window, mainly in theoretical contexts. However, in practical coding, the $\theta$ term was eventually removed through formula adjustments. Progressing from S4 (ICLR 2022), state space models began incorporating a trainable discrete step size $\Delta$ as a dynamic measurement window $\theta$, with additional insights provided in Figure 3 (a). **In Remark 3.4, we say $\Delta$ as an attention mechanism.** This terminology is also used in HTTYH (ICLR 2023) and Mamba (ICML 2024). By structuring $\Delta$ as (B, L, D), where each time step of the input signal has a unique discrete step size, the model can adaptively assign different weights to information at each step, revealing important sequence elements. This behavior resembles an attention mechanism (a gating mechanism) where attention scores are computed not through pairwise products but via a linear layer directly derived from the data itself (Line 120).
> * **Q7: Details about $\Delta B$**: Given $B$ (B,L,N), $\Delta$ (B,L,D), $\Delta B$ is in shape (B,L,D,N). The corresponding code is **DeltaB = Delta.unsqueeze(-1) * B.unsqueeze(2)**, which means (B,L,D,1)*(B,L,1,N).
> * **Q8: Details about $H$**: $H$ denotes the hierarchical projection matrix we have defined. As detailed in line 136, the state space model allows us to estimate the dynamical trajectory using a series of $\theta /\Delta$. However, attractor structures exhibit diverse forms (points, loops, surfaces), necessitating multiple scales of $\theta (e.g., 2\theta, 4\theta, 8\theta)$ for accurate approximation. To address this, as illustrated in Figure 3 (b), we iteratively combine $\theta$ in powers of 2 (repeatedly folding the sequence to acquire representations at varied scales). The matrix $H$ signifies the projection from the prior scale to the subsequent scale. Based on the distinctions between the left and right $\theta$ intervals, it can be further segmented into $H^{\theta1}$ and $H^{\theta2}$. These concepts encompass piecewise polynomial approximation and wavelet transforms (line 466).
>
> * **W2: About Chaos System:** Chaos, a characteristic of nonlinear dynamical systems, has been a focal point of extensive study. The Takens and Whitney theorems serve as valuable tools for exploring chaos. Following your recommendation, we will change **Chaos perspective** to **Dynamic systems perspective** within the document.
>
> * **W1,W3,W4**: Paper Presentation:** We trust that the responses to **questions 1-8** have assisted in clarifying any uncertainties related to symbol interpretations. In the revised manuscript, we will rephrase sections related to the Takens theorem and integrate a more thorough background on state space models to ensure accessibility for readers across various disciplines.
>
> Considering reviewers zUbZ/zqBV, all agreed that our paper has good merits such as **satisfactory novelty**, **good evaluation**, and **efficiency**, we believe our research findings are worth sharing with the research community. If your queries have been addressed, we kindly request an increase in the rating and confidence level.

---

> > ### Comment · Reviewer_qpQ4 · 2024-08-08
> >
> > Thanks for the clarification. Hope the authors will polish their writing in the final version.

---

> > > ### Author Response · Authors · 2024-08-10
> > >
> > > Thank you for your feedback and validation of our paper! We greatly appreciate your help in improving our scores! Wishing you all the best!

---

### Official Review · Reviewer_zqBV · 2024-07-15

**Soundness:** 3
**Presentation:** 2
**Contribution:** 3
**Rating:** 7
**Confidence:** 3

**Summary:**

The paper introduces chaos theory into a long-term time series forcasting (LTSF) model called Attraos (a play on the words "attractor" and "chaos"). They propose a Multi-resolution Dynamic Memory Unit (MDMU) which is inspired by (and looks a lot like) the State Space Models (SSMs) used in the Mamba family of models. However, unlike SSMs, Attraos assumes additional structure on top of time-series signals by utilizing attractor behavior from chaos theory in the phase space. Attraos outperforms many existing architectures (e.g., Mamba, RWKV-TS, PatchTST) on time-series tasks at a fraction of the computational complexity and cost.

**Strengths:**

- S1: **Outperforms existing architectures at a fraction of the cost**. Attraos is able to outperform models like Mamba, RWKV, and Transformers on many different time series tasks at a fraction of the computational cost (in terms of training time and parameter count).
- S2: **Theoretically complete**. The paper provide rigorous definitions, theorems, and proofs throughout the main paper and appendix to justify their method. (Note: before this paper I was quite unfamiliar with chaos theory, though I have some experience in SSMs and time-series. I am unable to verify much of the math.)
- S3: **Comprehensive empirical results**. The paper compares Attraos to many alternative architectures for time-series data (e.g., RWKV, Mamba, Transformers, MLPs, etc.) and shows that on many tasks Attraos outperforms alternative methods at a fraction of the computational cost. Additionally, the paper includes an ablation study for the different components of Attraos.

**Weaknesses:**

- W1: **Presentation of results could be clearer**. I was often confused by the choice of coloring in the tables. See the following for my suggestions:
    1. [Table 1,2,4]. The "Red/blue" colorscheme is unfortunate, because most brains (including mine) associate the color "red" with "poor performance". I suggest using **bold** for best, <u>underline</u> for second best, as is often done in AI papers.
    2. [Table 3,5]. For "improvement/decline" in performance, I suggest one of two options: (1) show "better performance" with blue, and "worse performance" with bold and red to emphasize decreased performance from an ablation, or (2) showing only the "deltas" with a "stock ticker arrow" (🔺 or 🔻, colored green for improvement, red for decline).
- W2: **Complete architectural description is missing**. Most of the paper focuses on describing how to implement the special components of Attraos, but the complete picture of the architecture is missing. Additionally, there is no section describing the hyperparameters for the training setup for each experiment. See Q1 for specific questions.
- W3: **Missing error bars**. Table 1 represents average results of long term forecasting across a swath of model classes, but these averages are not accompanied by reports of standard deviation. Thus, it is difficult to tell whether improved average performance is actually signficant. Improve by including error bars for all numerical results, which may require re-running some experiments under different random seeds.
- W4: **Grandiose, non-academic language**. The paper starts with "In the intricate dance of time" [L15] 😂 and makes statements like "we can transcend the limitations of deterministic dynamical systems" [L34]. I suggest reworking these lines to preserve the professionalism of the rest of the work.

These weakenesses are admittedly small and easily fixable by the authors during the review process. My overall attitude towards the paper is that it is of high quality and should be accepted. However, a lot of the theory was beyond my ability to evaluate, and I am willing to adjust my score positively/negatively as I become more familiar with this work throughout the review process.

**Questions:**

- Q1: The complete architecture for Attraos is not described. Do you just take Mamba and modify the SSM component? What is the complete set of learnable parameters? Where does the Hopfield Network come into play and how are its memories trained? What are the training hyperparameters for each experiment?

**Limitations:**

Limitations are adequately described in the conclusion.

---

> ### Author Rebuttal · Authors · 2024-08-07
>
> We sincerely thank you for appreciating the theoretical design of our model and its efficiency. We apologize for missing several details and would like to clarify as follows:
>
> * **W1: Unclear presentation of results:** Your comment is really helpful. We have followed your comments to indicate the best/worst performance in our revision. Thank you.
>
> * **W2: Lack of several architectural details:** Thank you for pointing out this issue. Please see our reply in Q1.
>
> * **W3: Missing error bars:** Thank you for your correction. Subsequently, we chose 3 random seeds for experimentation. Due to time constraints, we have provided the standard deviation results （MSE）for select datasets. The comprehensive experimental error data and associated error bars will be included in the Appendix.
> ||ETTh1|ETTm2|Weather|
> |---|---|---|---|
> |96|0.002|0.003|0.003|
> |192|0.003|0.003|0.001|
> |336|0.001|0.002|0.001|
> |720|0.005|0.003|0.002|
>
>
>
> * **W4: Several non-academic words**: We have revised this part based on your advice. Thanks for raising this concern.
>
> * **Q1.1: The complete architecture of Attraos**: We have streamlined the architecture by removing redundant modules from Mamba (such as gated multiplication and local convolution), keeping only the fundamental SSM kernel $K(t,s)=e^{tA}B$. This refinement enhances interpretability and recent studies suggest that redundant modules can introduce side effects in time series prediction tasks (https://arxiv.org/pdf/2406.04320, https://arxiv.org/pdf/2405.16312). To facilitate a deeper grasp of the Attraos framework, we present pseudocode below for your enhanced understanding, and a model pipeline structure diagram will be added in the appendix:
>
> | Input | Discrete historical data {z} in batch, phase space reconstruction hyperparameters {m, τ} |
> | --- | --- |
> |Initialize State Space Model| Obtain $B$ (B,L,N), $\Delta$ (B,L,D), $H$ (B,L,N) through three linear layers, with special initialization of $A$ (D,N)|
> |Discretize State Space Model| $\overline{A} = e^{\Delta A}$ (B,L,D,N), $\overline{B} = \Delta B$ (B,L,D,N) |
> |1|Reconstruct dynamical trajectories in phase space (Equation 5) and obtain patch representation $u$ (B,L,D) (Equation 6)|
> |2| Improved Blelloch scan algorithm, obtaining dynamical system representation $x$ (B,L,D,N) through $u$, $H$, $\overline{A}$, $\overline{B}$ (line 205) |
> |3| Frequency domain evolution: $\mathcal{F}\circ\mathcal{K}\circ x \circ\mathcal{F}^{-1}$|
> |4| Flatten Tokens, obtaining prediction results through observation function $W_{h}$|
> | Output | Discrete future prediction data |
>
>  * **Q1.2: Details about Hopfield Evolution**:
> The Hopfield network evolution strategy is one of the three mainstream attractor evolution methods mentioned in the paper (line 178). It is positioned alongside frequency domain evolution and direct evolution strategies, used for system evolution on the dynamic representation built on multi-resolution dynamic memory units (MDMU). A detailed introduction to the Hopfield network is provided in **Appendix A2 on page 13**, and the modern version of the Hopfield network(https://arxiv.org/pdf/2008.02217) is essentially a cross-attention mechanism. By employing predefined trainable tokens as attractor memory banks (Key and Value in attention), and using data as queries, the overall energy function of the system is minimized during model training to achieve stability. The trainable tokens in the stable state can be considered as the system's attractor memory.
>
>  * **Q1.3: Experimental hyperparameters**:
> The experimental hyperparameters can be located in **Appendix D3 on page 21**. For further details concerning SSM and mathematical aspects, please refer to my response to the third reviewer (qpQ4).
>
> In the revision, we have carefully incorporated your comments in the revised paper. Considering reviewers zUbZ/qpQ4, all agreed that our paper has good merits such as **satisfactory novelty**, **good evaluation**, and **efficiency**, we believe our research findings are worth sharing with the research community.

---

> > ### Comment · Reviewer_zqBV · 2024-08-08
> >
> > In this year's review cycle I am unable to see your paper's revisions, but I trust the authors that my suggested changes have been made. Similarly, I thank the authors for answering my questions. After reading the other reviews and responses, I see no reason to decrease my score and keep my vote to accept this paper.

---

> > > ### Author Response · Authors · 2024-08-10
> > >
> > > Thank you for your response! We greatly appreciate your support in recommending our paper!  Wishing you all the best!

---

### Official Review · Reviewer_zUbZ · 2024-07-24

**Soundness:** 3
**Presentation:** 2
**Contribution:** 3
**Rating:** 6
**Confidence:** 3

**Summary:**

The paper introduces a novel approach, named Attraos, for Long-term Time Series Forecasting (LTSF) based on treating the observed time series as high dimensional chaotic dynamical system. The model first estimates an embedding of the data through Phase Space Reconstruction (Takens embedding) and then utilizes a memory unit through state space model that represents the dynamical system through polynomials which allows to evolve the dynamics into future steps and forecast the data. The model is evaluated on multiple LTSF datasets and additional experiments such as ablations and robustness under noise are performed.

**Strengths:**

S1. The approach is proposing to forecast data evolution through learning the dynamical system that can generate such time series vs. to predict directly from data.

S2. Novel approach for polynomial estimation through a state space model is proposed.

S3. Fundamental properties of the approach are rigorously shown.

S4. Incorporation of Belloch algorithm for speed up is used.

S5. Experiments show that the approach performs better in many cases than existing approaches and ablation experiments are performed.

**Weaknesses:**

W1. There seems to be a detachment between the theorems and properties proved and the proposed system. There seems to be no discussion about how these properties lead to the particular setup in the paper. For example why this particular SSM was used? How the hyperparameters were chosen in light of the propositions? What are the cases that the model is limited and not expected to be effective? Also interpretation of the results is lacking.

W2. For non toy experimental results exposition and interpretation of the resulting dynamical system is missing. What is the dynamical system/s that are being obtained for the benchmark and what is the distribution of polynomials?

**Questions:**

See questions in W1 and W2 and also:

Q1. The Phase Space Reduction approach seems to be the well known Takens embedding and authors mention Takens thm, however call the method differently. Is there something different that I'm missing in PSR that is not in the classical computation of Takens embedding?

Q2. With local evolution I assumed that authors mean single step "autoregressive" forecasting. Is that the case? How would that change for multi-step?

Q3. It would be informative if the benefit/limitation/motivation of frequency evol vs. direct evol be more elaborately explained.

**Limitations:**

See W1-W2 and Q1-Q3. More extended discussion of limitations would contribute to better evaluation of the work vs generalization.

---

> ### Author Rebuttal · Authors · 2024-08-07
>
> Thank you very much for appreciating our technical novelty and the SOTA performance achieved by our method. We are really sorry for missing several details. Here we endeavor to address your questions.
>
> * **W1: Lack of discussion on model settings**:
>
> **[why using this particular SSM]:** The paper opts for an $A$ matrix structured as {-1, -1, -1...}, considering it a rough approximation of $Hippo-Leg$, which relies on finite measure window approximations utilizing Legendre polynomials (Line 478), aligning with our objective of approximating dynamical systems using windows $[t, t+\theta]$.
>
> **[how the hyperparams are chosen]:** In the realm of SSM-related investigations, diverse $A$ matrices exhibit varying characteristics; for example, a diagonal matrix like {-1, -2, -3...} might emphasize past information, while a step matrix could function as a localized attention mechanism, treating specific time steps collectively. Table 3 illustrates the performance stability enhancement of our designated $A$ matrix compared to randomly initialized $A$ matrices. The revised article will encompass a more extensive variant analysis.
>
> **[Limitation]:** Irrespective of the chosen $A$ matrix, it's crucial that its diagonal elements are negative (following left-half plane control theory) to avert gradient explosions. Moreover, the diagonal state space may limit the expressive power of the model. This paper partially addresses this issue by introducing multiple orthogonal subspaces (MDMU block).
>
> * **W2: Missing additional analysis.**:
>
> **[non-toy experiment]:** In Appendix E, we have delved into Chaos Modulation (E4: rectifying trajectories with real values throughout model evolution), Chaos Reconstruction (E5: visualizing the constructed model's dynamical system alongside the actual dynamical system), and Chaos Representation (E6: exploring the impacts of truncating and non-truncating trajectories). Should you necessitate additional detailed analysis, kindly inform us, and we will promptly address your queries.
>
> **[the dynamics of other baselines]** As this paper is the first to restore the underlying dynamic structure in a time-series prediction benchmark using PSR technology and innovatively utilize polynomial estimation, whereas other baseline models are purely data-driven, it is challenging to access the dynamic systems built by other models. Detailed explanations regarding the real dynamical systems behind the datasets are provided in Appendix E1 and E2.
>
> **[Polynomial distribution]** Given that the state-space model is initialized from specific polynomials and then adapted through open matrix gradients, making it challenging to access the specific polynomial distributions behind the model. In recent work, the state-space model has been defined to adaptively update to the most suitable polynomial space for modeling the dynamics of the time series through gradient updates. (https://arxiv.org/pdf/2405.16312)
>
> * **Q1: Notation of PSR**: Takens embedding often refers to phase space reconstruction, while Takens theorem is broader, ensuring attractor structures can be recovered even in dimensions greater than twice the original sequence. These are roughly equivalent terms without a unified label in dynamic literature.
>
> * **Q2: Local Evolution**: Local evolution is not actually a single-step autoregression. In lines 81-83, the adjacent dynamic trajectory points belonging to the same attractor will undergo evolution using the same evolution operator $K_i$ (for example, time steps {1, 7, 8, 11} use the $K_1$ operator, and time steps {2, 5, 67, 68} use the $K_2$ operator). In the direct evolution strategy, we employ the KNN clustering algorithm to partition the points belonging to each attractor based on the Euclidean distance in phase space. In the frequency domain evolution strategy, we directly use dominant modes in the frequency domain to represent attractors.
>
> * **Q3: Frequency Evolution**: **[Motivation]:** We chose the frequency domain evolution strategy inspired by articles in the field of neuroscience, where attractors amplify in the frequency domain (lines 54-56).
> **[Limitations]:** Regarding interpretability, the frequency domain evolution strategy is less effective than direct evolution (dividing attractors based on Euclidean distance clustering).
> **[Advantages]:** Direct clustering can lead to incorrect attractor divisions due to the significant noise present in real-time series data. Frequency domain evolution, on the other hand, filters out some high-frequency noise, enhancing the model's robustness and consequently improving its performance.
>
> In the revision, we have carefully incorporated your comments in the revised paper. Considering reviewers zqBV/qpQ4, all agreed that our paper has good merits such
> as **satisfactory novelty**, **good evaluation**, and **efficiency**, we believe our research findings are worth sharing with the research community. We sincerely hope that a revision is still considered.

---

> > ### Author Response · Authors · 2024-08-12
> > **Kindly Request for Reviewer's Feedback**
> >
> > Dear Reviewer zUbZ,
> >
> > Since the End of author/reviewer discussions is coming in one day, may we know if our response addresses your main concerns? If so, we kindly ask for your reconsideration of the score. Should you have any further advice on the paper and/or our rebuttal, please let us know and we will be more than happy to engage in more discussion and paper improvements.
> >
> > Thank you so much for devoting time to improving our paper!

---

> > ### Comment · Reviewer_zUbZ · 2024-08-13
> > **Authors Rebuttal Response**
> >
> > I would like to thank the authors for their concise yet informative rebuttal that have clarified most of my questions. After reading authors rebuttal to my review and other reviewers I remain positive and supportive of the work and novelty of the approach.
> >
> > One issue that I am still wondering about and would like to know if authors are able to further clarify is the detachment of the theory and the approach choices. Could the authors summarize the outcomes of the properties developed in the work and how they led/constrained model choices?

---

> > > ### Author Response · Authors · 2024-08-13
> > >
> > > Thank you for your response and support! We would be delighted to provide you with further clarification on the theory and approach choices.
> > >
> > > * **Motivation and Significance**: Machine learning has historically been bifurcated into physics-driven approaches (such as PDE numerical solvers, PINN, etc.) and data-driven methods (like large models, various neural networks, neural operators, etc.). While data-driven techniques have excelled in time series modeling (as well as in computer vision, natural language processing, etc.), their black-box nature poses challenges for sustained advancement within the machine learning community. Recent publications in esteemed journals like Nature, Science, PNAS, etc., have showcased a fusion of physics-driven and data-driven methodologies, hinting at the emergence of a prominent research trend involving the integration of physics priors into deep learning models. **Our research marks the inaugural incorporation of PDE dynamical system knowledge into the realm of time series prediction, promising to facilitate interdisciplinary amalgamation and the creation of interpretable deep time series models.**
> > > * **Dynamic System**: also known as PDE dynamical systems. Let the domain $S$ be an open subset of $\mathbb{R}^d$ and set an integer $k \geq 1$. Define the system state as $\boldsymbol{x}: S \mapsto \mathbb{R}^m$ where $\boldsymbol{x}=\left(x^1, \ldots, x^m\right)$. Then, an expression of the form:$\mathcal{F}\left(D^k \boldsymbol{x}(s), D^{k-1} \boldsymbol{x}(s), \ldots, D \boldsymbol{x}(s), \boldsymbol{x}(s), s\right)=0$ is called a $k^{\text {th }}$-order system of partial differential equation, where $\mathcal{F}: \mathbb{R}^{m d^k} \times \mathbb{R}^{m d^{k-1}} \times \ldots \times \mathbb{R}^{m d} \times \mathbb{R}^m \times S \mapsto \mathbb{R}^m$ and $s \in S$.
> > >
> > > real-time series are observed values obtained from the system through a function h. By starting from dynamical systems instead of the original value, it becomes possible to have a better understanding of the fundamental temporal behavior. Moreover, this approach offers the advantages of interpretability and visualization.
> > >
> > > * **Chaos Theory**: The study of chaos can be considered a branch of dynamical systems, focusing on the fact that both linear and nonlinear dynamical systems tend to exhibit certain fixed shapes in their trajectories, known as attractors. What may appear as irregular behavior in the time domain often reveals stable structures in the dynamical trajectories. The concept of attractors can readily be correlated with current deep-learning pattern recognition technologies.
> > >
> > > * **Why SSM?**: The reason for choosing SSM to encode dynamical structures is that polynomials are widely recognized as universal approximators in dynamical systems research. The mathematical interpretation of SSM conveniently aligns with polynomial projection, making SSM well-suited for encoding dynamical structures. And we have introduced a novel SSM evolution matrix {-1, -1, -1, ...} to describe finite window approximations.
> > >
> > > * **Why SSM + Multiple Orthogonal Subspaces**: Building upon SSM, we drew inspiration from chaos-related research (**Theorem 3 Attractor evolution error, line 163**) and proposed an enhanced version of SSM that utilizes distinct orthogonal subspaces to store various attractor structures. This innovation was validated through experimental results.
> > >
> > > * **Why Frequency Evolution**: In the field of neuroscience research (where EEG and ECG are recognized as chaos datasets), it has been established that attractors are amplified in the frequency domain. Therefore, this paper adopts a frequency-domain evolution strategy. Besides, by discarding high-frequency components, noise effects can be mitigated, reducing computational complexity. Additionally, we explored the effects of Hopfield evolution and KNN clustering evolution, and through experimentation, **we found that these two methods were not as effective as frequency-domain evolution. We provided explanations for these findings in line 257**.
> > >
> > > * **Follow-up Work**: In our recent research, we have discovered that **incorporating dynamical system priors leads to performance enhancements across all four types of time-series tasks (prediction, classification, interpolation, anomaly detection) and all four neural network architectures (convolutional, attention-based, linear, SSM). These improvements include a tenfold reduction in parameter count, as well as more stable gradients.** This work is set to be released soon—stay tuned for more updates.

---

### Author Rebuttal · Authors · 2024-08-07

We commence by thanking the reviewers for their insightful comments. We are pleased to see that all the reviewers agree with some strengths of our paper, such as technique novelty(**Reviewer zUbZ, zqBV, qpQ4**), comprehensive evaluation (**Reviewer zUbZ, zqBV, qpQ4**), and efficiency (**Reviewer zUbZ, zqBV, qpQ4**). Your expertise also helpsus to strengthen our paper significantly.

We apologize for any inconvenience caused by the omission of certain details in the article and endeavor to respond to each comment. We sincerely hope that the responses can release the reviewers' concerns. For reference, we present a brief introduction of the response as follows.

### **In response to reviewer zUbZ:**
* We extensively examined the **SSM setting**, including the reason for choosing the {-1, -1, -1...} form of the A matrix, its appropriate applications, and practical constraints.
* We provided a further explanation on **local evolution**.
* We extensively examined the reasons, advantages, and limitations of the **frequency evolution strategy**.

### **In response to reviewer zqBV:**
* We provided the standard deviation (**error bar**) of the experimental results on several datasets.
* We presented a **pseudocode** for the model to aid in a better understanding of the architecture.
* We offered a further explanation on how the **Hopfield network stores attractor memories** during the training process.

### **In response to reviewer qpQ4:**
* We offered detailed explanations on 8 specific questions regarding the **SSM kernel and computational details**.

Given that **all reviewers have rated soundness and contribution as 3 (good)**, we will strive to enhance the presentation of this paper. We firmly believe that our novel approach/initiative can offer the community fresh perspectives and technical contributions.

---

### Decision · Program_Chairs · 2024-09-25

**Decision:**

Accept (poster)

**Comment:**

The paper presents a novel model for time-series forecasting building on a chaotic dynamical systems view: the approach is original and robust, both empirically and theoretically. This has been widely recognized by the Reviewers.
Issues were mainly associated with presentation aspects and they have been appropriately addressed and clarified through the rebuttal and the discussion period.
The work can therefore be accepted pending that the Authors thoroughly revise and rewrite Section A.1 which appears to be a verbatim (and unacknowledged) copy of the Wikipedia page on Takens' theorem.